# WATERMARKING FOR USER IDENTIFICATION IN LARGE LANGUAGE MODELS

## ABSTRACT

We identify a new task for watermarking – namely the simultaneous identification of text as being automatically generated alongside the identification of the LLM user. We show that a naïve approach that treats a text as artificially generated if a user is correctly identified is prone to problems of false positives arising from multiple hypothesis comparison. We propose a novel approach [1] that is much more robust to the number of users. We derive theoretical bounds, and empically validate our approach.

## 1 INTRODUCTION

Large Language Models (LLMs) (Radford et al., 2018; 2019; Brown et al., 2020; Achiam et al., 2023; Touvron et al., 2023a;b) have emerged as the dominant technology across a wide range of natural language processing tasks. While these models have significantly advanced the field, their misuse has raised numerous ethical concerns. Of particular concern is the use of LLMs to imperson-ate human text, and to generate text that appears to represent a sincere effort to engage, but is in fact automatically generated. Example use cases include the automatic generation of homework; of sci-entific papers, grants, and reviews; and the automatic generation of spam and astroturfing (Wachter et al., 2024; Yang et al., 2023; Nikiforovskaya et al., 2020). The primary concern here is that such automatically generated text can flood the environment, making it impossible to identify sincere texts that are worth reading and responding to.

In response to these harms, researchers have suggested the use of digital watermarking (Aaronson & Kirchner, 2023; Fernandez et al., 2023; Kirchenbauer et al., 2023b). These approaches modify the sampling step of the LLM, allowing for the automatic identification of texts generated by a particular sampler. If the use of watermarked samplers becomes common, it would be much easier to identify and manage LLM generated content. However, this is not the only use of digital watermarking. As experimentally shown in Fernandez et al. (2023), digital watermarking can also be used to identify individual users of an LLM.

This use of watermarking, which we refer to as *user watermarking*, caries with it the risk of privacy violations. For example, if LLMs are used as part of a pseudoanonymization process to rephrase text, user watermarking could allow the text to be traced back to the account that rephrased the text. However, if appropriately disclosed, it could replace more intrusive governance measures. For example, user watermarking would allow LLM service providers to identify and shut down accounts used for generating spam without requiring the monitoring of every customer's API call.

However, user watermarking cannot be simultaneously performed alongside standard watermarking (i.e. identifying if a text is LLM generated). While the most obvious option is to simply say that a text is LLM generated if it is consistent with any user watermark – this is a form of multiple hypothesis testing (i.e. either the text was generated with user 1, or user 2, or ...), and if the test is fixed at a particular sensitivity, we find that the number of false positives increases as the number of users considered grows, a phenomenon we refer to as the false recognition problem. As discussed in the research by Liu et al. (2023b); Giboulot & Teddy (2024), false positives are considerably more critical than false negatives, as erroneously identifying human-generated texts as watermarked can result in more severe adverse consequences, including the wrongful suspension or shutdown of user accounts. We demonstrate, both theoretically and empirically, that as the user size expands, the false recognition bound increases exponentially. This rapid increase leads to failed watermark

---

[1]Our code is submitted as supplementary material. We will opensource it on Github after the anonymity period.

detection, even within a relatively small key space, hindering the watermark's capacity to embed richer information.

To address this issue, we propose a novel Dual Watermark Inspector (DWI) designed to detect the information embedded within the watermark while mitigating the false positive recognition problem. Our approach involves a joint encoding of indicator variable and key information into the text. Specifically, the indicator variable is a binary variable that signals if the text is watermarked. In contrast, the key information contains user identity encoded as an integer key, following the approach of previous work (Fernandez et al., 2023). This approach can be easily extended to accommodate more complex types of information. We use a hash decision function to determine whether the current token encodes indicator or key information. Then, we follow Aaronson & Kirchner (2023); Fernandez et al. (2023) to use the Gumbel trick to generate the current token. Afterwards, we calculate statistics for the indicator variable and key information respectively, mark the text with high indicator score as watermarked, and return the key with the maximal score as the encoded key.

Furthermore, we perform a rigorous analyses of the false positive error bounds associated with traditional methods compared to our proposed method. We prove a false positive error bound of a standard threshold method (Fernandez et al., 2023) as well as Multibit Yoo et al. (2023b). Our theory shows that the False Positive Ratio (FPR) can grow uncontrollably for threshold-based methods (We refer to this as Full Key Encoding (FKE) in this paper) particularly when the key space, corresponding to the number of user identities, is large or the text length is short. We analyze our method theoretically and demonstrate that it significantly outperforms the FKE method. Specifically, we prove that if the key space is larger than around 100, our method has a better FPR upper bound than FKE. Additionally, we propose a Hybrid Dual Watermark Inspector (HDWI) strategy that combines the DWI method and the original FKE method, with another theoretical bound to guarantee performance. Both our theoretical and empirical findings indicate that our proposed approach surpasses existing methods and substantially decreases the false recognition rate.

The contributions of this paper are as follows: (1) We conduct a rigorous theoretical and empirical analysis of the false recognition problem in watermarking for LLMs; (2) We propose a novel analysis of the false positive rate, and illustrate the potential severity of this problem under specific conditions, such as variations in text length, key space size, and other relevant factors; (3) We introduce a novel method DWI to reduce the false recognition rate, while preserving a high true positive rate, and empirically demonstrate the effectiveness of our proposed method.

## 2 RELATED WORKS

We categorize existing watermarking methods into two categories: Watermark Indicators and Information Conveyance. Watermark Indicators encode a specific indicator within LLM-generated text, that indicates whether the text is generated by a watermarked LLM or not. Aaronson & Kirchner (2023); Fernandez et al. (2023); Fu et al. (2024) suggest employing the Gumbel trick to generate a corresponding random variable with a distinct distribution for watermarked text. Kirchenbauer et al. (2023a;b) propose dividing the vocabulary into red and green lists based on preceding tokens. Christ et al. (2024) introduce the concept of embedding undetectable watermarks in language model outputs using cryptographic techniques. Zhao et al. (2023) propose the Unigram-Watermark method to improve the detection accuracy and robustness of watermarks in LLM-generated text.

On the other hand, Information Conveyance methods embed much richer information within the generated text. Most existing methods embed an integer key into the generated text, which could be used to represent user ID. Fernandez et al. (2023) propose utilizing a hash key to represent essential information. Yoo et al. (2023a;b); Wang et al. (2023); Boroujeny et al. (2024); Qu et al. (2024) suggest encoding multibit information in the watermark. Abdelnabi & Fritz (2021) advocate for adversarially encoding information into the watermark. However, these approaches determine whether the text is watermarked by reusing the key information. To the best of our knowledge, no prior work has analyzed how such reusing may result in false recognition as key capacity increases.

In general, the distinction between Indicator and Information Conveyancing is somewhat artificial. Most indicator methods Aaronson & Kirchner (2023); Fu et al. (2024) draw on work on cryptographically secure hashing, and their behaviour depends on a hidden salt. Instead of randomly selecting salts, they can simply be chosen to encode multibit data, and this approach was used by (Fernandez et al., 2023) to encode user IDs.

## 3 DUAL WATERMARK INSPECTOR

We propose a novel approach termed the Dual Watermark Inspector (DWI) to address the false recognition problem when embedding keys with capacity $K$ into LLM watermarking. Here, capacity $K$ refers to $K$ distinct keys that can be encoded, such as a user ID. Instead of relying on all tokens to encode the key information, we selectively require certain tokens to encode the indicator as well as the key information. Additionally, we propose a Hybrid Dual Watermark Inspector (HDWI) method that further leverages the key information to assist in detecting the watermark.

Our work builds upon Aaronson & Kirchner (2023); Fernandez et al. (2023), who deliberately constructed a statistic to guide both the generation and detection strategies. Although the original paper does not explicitly identify this method as the Gumbel-Max trick (Gumbel, 1954; Maddison et al., 2014; Jang et al., 2016), we show that this method is essentially a Gumbel-Max trick (Section 3.3).We offer additional details to enhance the understanding of how and why the Gumbel-Max trick contributes to effective watermark detection. This formulation underpins our analysis in Section 4. In our discussion, we refer to the method introduced by Fernandez et al. (2023) as Full Key Encoding (FKE), where all tokens are used to encode a single key.

**Notation.** We denote the key ID by $\xi \in [1, K]$, with $K \in \mathbb{N}$ representing the total number of keys i.e. the capacity. The token sequence $[x_1, \ldots, x_T]$ is generated by an LLM $\mathcal{L}$, where each token $x_i$ is within the range $[1, \ldots, V]$, $V$ being the vocabulary size. Probabilities for predicting the next token are denoted as $p_i$, with corresponding logits $\ell_i$, which are adjusted to $\tilde{\ell}_i$ by incorporating the Gumbel variable. The indicator function $\mathbb{1}(\cdot)$ returns 1 if the condition is true and 0 otherwise. The hash function $\mathcal{H}$ is used to calculate the hash key. A uniform distribution $U(0, 1, s)$ is used, where $s$ is the random seed to generate the standard uniform random variables. The scores $S_d$ and $S_k(\xi)$ are used to detect the presence of watermarks and to retrieve key information, respectively. The parameter $r$ controls the proportion of tokens for encoding the indicator variable, while $\tau_k$ and $\tau_d$ are threshold variables, respectively, used to determine if text contains a watermark, and or a particular key.

### 3.1 GENERATING

During generation, we follow the approach outlined by Aaronson & Kirchner (2023); Fernandez et al. (2023), using a large language model (LLM) to generate a text sequence while encoding specific information through a deliberately designed sampling strategy. This strategy modifies standard stochastic sampling (simply sampling a token based on the corresponding probability) by incorporating the Gumbel-Max trick, with the random seed for the Gumbel random variable being controlled by the previous tokens and the information to be encoded. We discuss the relationship between the Gumbel-Max trick and the original method (Aaronson & Kirchner, 2023; Fernandez et al., 2023) in Section 3.3. The algorithm outline is provided in Algorithm 1.

Given a key ID $\xi$ and a token sequence $[x_1, \ldots, x_{i-1}]$, where each $x_j$ is a token ID within the range $[1, \ldots, V]$, with $V$ representing the vocabulary size, an LLM $\mathcal{L}$ generates the subsequent token. This is done by calculating the logit as $\ell_i = \mathcal{L}([x_1, \ldots, x_{i-1}])$, where $\ell_i \in \mathbb{R}^V$ represents the logits for predicting the next token. In the standard generation process, various sampling methods are used to select a token based on this logits vector, including Top-k sampling (Fan et al., 2018), Nucleus sampling (Holtzman et al., 2020), and Stochastic sampling (Fan et al., 2018; Holtzman et al., 2020; Fu et al., 2021). In this paper, following Aaronson & Kirchner (2023); Fernandez et al. (2023), we focus on stochastic sampling to direct sample a token based on the corresponding probability, which can be easily extended to other sampling methods, as discussed in Fernandez et al. (2023).

To embed both the indicator and the key information into the watermark, we first use an indicator hash key $h_i$ to determine whether the current token encodes the indicator variable or key information. Specifically, a hash key $h_i = \mathcal{H}(x_{i-h}, \cdots, x_{i-1})$ is calculated based on the previous $h$ tokens, and $d_i$ is determined using the indicator function $d_i = \mathbb{1}((h_i \bmod 100) < 100 r_d)$, where $h_i \in \mathbb{N}$ is the hash key derived from $x_{i-h}$ to $x_{i-1}$. The indication ratio parameter $r_d \in [0, 1]$ is a user-specified ratio that controls the proportion of tokens used for the indicator variable. The value $d_i$ can be either 0 or 1: if $d_i = 0$, it indicates that the current token will encode an indicator to signify whether the text is watermarked; if $d_i = 1$, the token will encode the key information. The information salt key $A_i$ is then computed as: $A_i(\xi, d_i) = \xi \cdot \mathbb{1}(d_i = 1)$ where $\xi$ is the user-specified key information used to store keys such as a user ID. This encoding method is naturally robust to deletion or insertion, as the hash key depends solely on the previous $h$ tokens. If a small number of tokens are removed or added, most of the remaining salt keys remain unaffected, thereby preventing any significant change

to the final detection score. We also empirically illustrate this claim using the insertion and deletion attack experiments described in Appendix A.6.

Subsequently, a new random seed is generated using the hash function $h_g = \mathcal{H}(x_{i-h}, \cdots, x_{i-1}, A_i(\xi, d_i))$, which is then used to construct a standard uniform distribution $u_i \sim U(0, 1, h_g)$. Here, $u_i \in \mathbb{R}^V$ is a vector of standard uniform random variables generated with the random seed $h_g$. This uniform variable is transformed into a Gumbel variable vector $g_i = -\ln(-\ln(u_i))$, where $g_i \in \mathbb{R}^V$ is a vector of standard Gumbel variables with parameters $\mu = 0$ and $\beta = 1$. The adjusted logits are then calculated as $\tilde{\ell}_i = \ell_i + g_i$. By incorporating the Gumbel variable $g_i$, we change the sampling process typically used in LLMs to directly select the token with the maximum score of $\tilde{\ell}_i$. Consequently, the next token $x_i$ is determined as $x_i = \arg\max_j \tilde{\ell}_{ij}$, where $\tilde{\ell}_{ij}$ is the $j$th element of the vector $\tilde{\ell}_i$. The Gumbel trick ensures that the probability of sampling the $k$th token, $P(k = \arg\max_j \tilde{\ell}_{ij}) = p_{ik}$ is an unbiased estimator (Fernandez et al., 2023; Liu et al., 2023a), corresponds to the probability associated with the logit $\ell_{ij}$. This property guarantees that the sampling process remains an unbiased estimator of the original probability distribution. Thus, the sampling of the next token $x_i$ is now driven by a random uniform vector $u_i$, which is generated based on the previous tokens and the random seed $h_g$. Therefore, the essence of this conversion is that we shift from detecting the sampling process to performing statistical analysis on the random variables, which is more feasible and operational. When detecting, we analyze the distribution of the random variables to extract both the watermark indicator and key information embedded in the text.

---

**Algorithm 1** Watermarked Text Generation and Detection

**Generation Process:**
**Require:** Language model $\mathcal{L}$, key ID $\xi$, indication ratio $r_d \in [0, 1]$, token sequence $[x_1, \ldots, x_{i-1}]$
**Ensure:** Generated token $x_i$
1: Compute logits: $\ell_i = \mathcal{L}([x_1, \ldots, x_{i-1}])$
2: Compute hash key: $h_i = \mathcal{H}(x_{i-h}, \cdots, x_{i-1})$
3: Determine indicator: $d_i = 1$ if $(h_i \% 100) < (100 r_d)$ else $0$
4: Compute salt key: $A_i = \xi \cdot \mathbb{1}(d_i = 1)$
5: Compute random seed: $h_g = \mathcal{H}(x_{i-h}, \cdots, x_{i-1}, A_i)$
6: Generate uniform random variables: $u_i \sim U(0, 1, h_g)$
7: Transform to Gumbel variables: $g_i = -\ln(-\ln(u_i))$
8: Adjust logits: $\tilde{\ell}_i = \ell_i + g_i$
9: Next token: $x_i = \arg\max_j \tilde{\ell}_{ij}$
10: **return** $x_i$

**Detection Process:**
**Require:** Language model $\mathcal{L}$, token sequence $[x_1, \ldots, x_T]$, sequence length $T$, candidate keys $\{\xi\}$, ratio $r \in [0, 1]$, thresholds $\tau_d, \tau_k$
**Ensure:** Indicator variable $I_d$, key information $I_k$

1: Initialize indicator score: $S_d = 0$
2: Initialize key scores: $S_k(\xi) = 0$ for all $\xi$
3: **for** $i = 1$ to $T$ **do**
4:  Compute hash key: $h_i = \mathcal{H}(x_{i-h}, \cdots, x_{i-1})$
5:  Determine indicator: $d_i = \mathbb{1}((h_i \% 100) < 100 r_d)$
6:  **for** each candidate key $\xi$ in $\{\xi\}$ **do**
7:    Compute salt key: $A_i(\xi, d_i) = \xi \cdot \mathbb{1}(d_i = 1)$
8:    Compute random seed: $h_g(\xi) = \mathcal{H}(x_{i-2}, \cdots, x_{i-1}, A_i)$
9:    Generate uniform random variables: $u_i(\xi) \sim U(0, 1, h_g(\xi))$
10:   Update key score: $S_k(\xi) \mathrel{+}= d_i \cdot (-\ln(1 - u_{ix_i}(\xi)))$
11:  **end for**
12:  Update indicator score: $S_d \mathrel{+}= (1 - d_i) \cdot (-\ln(1 - u_{ix_i}(0)))$
13: **end for**
14: Compute indicator variable: $I_d = \mathbb{1}(S_d > \tau_d)$
15: Identify key information: $I_k = \arg\max_\xi S_k(\xi)$
16: **return** $I_d, I_k$

\* Green highlights represent newly added components in DWI compared with FKE.

---

## 3.2 DETECTING

When detecting the watermark (the algorithm outline is provided in Algorithm 1), we follow the same procedure by using the previous tokens and the salt key to recover the random variable $u_i(\xi) \in \mathbb{R}^V$ corresponding to the current token ID $x_i$ and a probing key ID $\xi$ and we denote $u_{ix_i}(\xi)$ as the $x_i$-th element of the vector $u_i(\xi)$. If the text is not generated by above generating procedure or if the salt key does not match, the corresponding random variables will simply be uniformly distributed. However, if the text is generated with the specified procedure and the correct salt key, the corresponding random variable will follow a Beta distribution (Fernandez et al., 2023), as it is the maximum element of a uniform vector. Then, we use a specific test variable $S_d$ to differentiate between these two distributions.

Specifically, given an LLM $\mathcal{L}$ and token sequence $[x_1, \ldots, x_T]$, where each $x_i$ is a token ID within the range $[1, \ldots, V]$ and $T \in \mathbb{N}$ is the sequence length, we detect the indicator variable $I_d \in \{0, 1\}$ and key information $I_k \in [1, \ldots, K]$ using deliberately designed score functions. Simliar to the generating phase, we first calculate the hash key based on previous tokens as $h_i = \mathcal{H}(x_{i-h}, \cdots, x_{i-1})$ and $d_i = \mathbb{1}((h_i \bmod 100) < 100 r_d)$, where $h_i$ is a natural number and $d_i$ is in $\{0, 1\}$. Unlike

the generating process, here, the key information is unknown and must be inferred, $A_i$ is made a function of $\xi$ and $d_i$, resulting in different values for $A_i(\xi, d_i)$ defined as $A_i(\xi, d_i) = \xi \cdot \mathbb{1}(d_i = 1)$.

Then, we calculate the random seed $h_g$ as a function of $A_i(\xi, d_i)$, denoted as $h_g(\xi) = \mathcal{H}(x_{i-h}, \cdots, x_{i-1}, A_i(\xi, d_i))$, and use it to generate the random seed $u_i(\xi) \sim U(0, 1, h_g(\xi))$, where $u_i(\xi) \in \mathbb{R}^V$ is a vector of standard uniform random variables generated with the seed $h_g(\xi)$. It is important to note that for unwatermarked text or when the salt key does not match the generating key, the $x_i$-th random variable in $u_i$ (denoted as $u_{ix_i}$) will simply follow a uniform distribution. However, if the text is watermarked and the salt key is correct, the $x_i$-th random variable in $u_i$ is sampled as the maximum of the Gumbel-modified logits $\ell_i$, leading to a Beta distribution (Fernandez et al., 2023). Similiar to the detection method (Aaronson & Kirchner, 2023; Fernandez et al., 2023), we calculate the score as $S_d = -\sum_i^T (1 - d_i) \ln(1 - u_{ix_i}(0))$ and $S_k(\xi) = -\sum_i^T d_i \ln(1 - u_{ix_i}(\xi))$, where $S_d$ is the indicator score to decide whether it is watermarked or not. Additionally, $S_k(\xi)$ is the key information score, indicating the likelihood that key $\xi$ is embedded. We calculate the indicator variable as $I_d = \mathbb{1}(S_d > \tau_d)$ and the user key information $I_k$ as the argument maximizing $S_k(\xi)$ which denotes as $I_k = \arg\max_\xi S_k(\xi)$.

The original paper showing user watermarking Fernandez et al. (2023) considered two separate tasks; either standard watermarking, in which all tokens are used to indicate if a text was LLM encoded, or a second strategy in which all tokens were used to encode key/user information. They do not consider the task of jointly identifying if it is LLM generated while simultaneously identifying a second key. To answer this question, we propose a simple baseline, FKE, that introduces a threshold $\tau_k$, where samples with the maximum score $\max_\xi S_k(\xi)$ below $\tau_k$ are considered not watermarked. In the FKE method, all tokens are used to encode the key information, and the maximal score is used to determine whether it is watermarked. We also explore extensions of the DWI framework to search for potential improvements. The FKE method and models we investigated are shown below:

**Full Key Encoding (FKE).** We extend the method proposed by Fernandez et al. (2023) by utilizing the maximal score within the full key space and checking if it exceeds a specified threshold $\tau_k$. In FKE, as all tokens are used to encode the key informaiton, $S_k(\xi) = -\sum_i^T \ln(1 - u_{ix_i}(\xi))$ . $I_d$ is determined as $I_d = \mathbb{1}(\max_\xi S_k(\xi) > \tau_k)$, where the condition evaluates whether the maximum score $\max_\xi S_k(\xi)$ surpasses the key threshold $\tau_k$.

**Partial Key Encoding (PKE).** To better compare the results of PKE with our proposed strategy, we utilize only a portion of the tokens to encode key information and use this information to determine whether the text is watermarked. The remaining tokens are left unused for encoding. In PKE, since only partial tokens are used to encode the key information, the score is calculated as $S'_k(\xi) = -\sum_i^T \mathbb{1}\{A_i(\xi, d_i) \neq 0\} \ln(1 - u_{ix_i}(\xi))$. The indicator $I_d$ is then determined as $I_d = \mathbb{1}(\max_\xi S'_k(\xi) > \tau_k)$. This method serves as an ablation study of the FKE method.

**Hybrid Dual Watermark Inspector (HDWI).** The HDWI method combines both the indicator variable and the key information to determine the watermark status, thereby reducing recognition errors. In HDWI, $I_d$ is calculated as $I_d = \mathbb{1}(S_d > \tau_d \cap \max_\xi S_k(\xi) > \tau_k)$, where the condition evaluates whether the score $S_d$ exceeds a certain threshold $\tau_d$ and whether the maximum score $\max_\xi S_k(\xi)$ surpasses the key threshold $\tau_k$.

**Mean Rebalance (MR).** This heuristic represents a natural variant of the HDWI technique, specifically designed to accommodate variations observed across different sequences. Since the mean value of scores can vary for each sequence, using a fixed threshold may lead to errors. To mitigate this, the MR method compares the maximum score $S_k(\xi)$ with the mean value of the scores and considers the sequence as unwatermarked if the difference is smaller than a particular threshold. The condition is adjusted as $I_d = \mathbb{1}(S_d > \tau_d \cap \max_\xi S_k(\xi) - \frac{1}{K}\sum_{\xi=1}^K S_k(\xi) > \tau_k)$.

**Second Rebalance (SR).** Similar to the MR method, the SR method utilizes the difference between the highest score and the second highest score in the sequence. The indicator $I_d$ can be calculated as $I_d = \mathbb{1}(S_d > \tau_d \cap \max_\xi S_k(\xi) - \max_{\xi \neq \arg\max S_k(\xi)} S_k(\xi) > \tau_k)$, where the condition specifies that the gap between the maximum score and the second largest score must exceed a threshold $\tau_k$.

## 3.3 GUMBEL-MAX TRICK EQUIVALENCE

It is worth noting that in Aaronson & Kirchner (2023)'s work, they do not explicitly generate the Gumbel variable and select the maximal one. Instead, they perform an equivalent trick by sampling $V$ random variables $u = (u_1, \ldots, u_V)$, where $u_v$ are i.i.d. with $u_v \sim U(0, 1)$. Then, given the

probability vector $p = (p_1, \ldots, p_V)$, the current token is selected as $V^\star = \arg\max_v u_v^{1/p_v}$. In Proposition 1, we demonstrate this method is essentially equivalent to the Gumbel-Max Trick.

# 4 THEORETICAL ANALYSIS

To provide a thorough understanding of the false recognition problem inherent in the traditional Full Key Encoding (FKE) method, and to demonstrate how our proposed DWI and HDWI approaches address this issue, we derive the false positive bound for these methods. We denote the capacity of key information as $K$ (which can be used to represent the total number of user IDs), the generated sequence length as $T$, and the user-specified threshold as $s$. We compute three false positive bounds. The first bound, presented in Theorem 1, provides a theoretical understanding of the baseline FKE method, which considers a document as watermarked if it matches any one of the key ID $\xi$. Next, we conduct the second analysis in Theorem 2 to demonstrate how and why our proposed DWI method effectively addresses the false recognition problem. Following this, we perform additional analysis in Theorem 3, illustrating how the HDWI method can further enhance performance. Then, we conduct a numerical experiment based on these bounds that empirically demonstrates the differences between FKE and our proposed approach DWI. We also present an theoretical analysis of multi-bit methods (Yoo et al., 2023b; Wang et al., 2023), based on Kirchenbauer et al. (2023a) in Appendix A.12, showing that false recognition persists as key capacity increases.

## 4.1 FALSE RECOGNITION BOUND

For FKE, we consider a text sequence to be watermarked if the maximum score $\max_\xi S_k(\xi)$ exceeds a certain threshold $\tau_k$. However, some unwatermarked text may also be mistakenly classified as watermarked due to an inherent issue with this method. Specifically, as discussed in Section 3.2, if the text is unwatermarked, all elements in the random variable vector $u_i(\xi) \in \mathbb{R}^V$, drawn during detection, will follow a uniform distribution. We denote $u_{ix_i}(\xi)$ as the $x_i$-th element of $u_i(\xi)$, which is the random variable corresponding to the generated token $x_i$. We can compute the probability that, given all $u_{ix_i}(\xi)$ are uniformly distributed, the statistic $S_k(\xi)$, constructed from $u_i(\xi)$, exceeds a specified threshold $\tau_k$. This probability corresponds to the false positive probability. We begin by establishing the following theorem.

**Theorem 1.** *Consider random variables $u_{ix_i}(\xi)$ drawn from a uniform distribution over $[0, 1]$, where $\xi = [1, \ldots, K]$ represents the key, and $K$ denotes the total key capacity. The index $i = [1, \ldots, T]$ refers to the $i$th token in the generated sequence. Each $u_i$ is a vector in $\mathbb{R}^V$, where $V$ is the vocabulary size, and $u_{ix_i}$ corresponds to the $x_i$th token in $u_i$. The score is calculated as: $S_k(\xi) = -\frac{1}{T} \sum_{i=1}^{T} \ln(1 - u_{ix_i}(\xi))$. We consider the sample is watermarked if $\max_\xi S_k(\xi) \geq \tau_k$, where $\tau_k$ is a threshold parameter. Then the false positive probability is bounded as follows:*

$$\Pr\left(\max_\xi S_k(\xi) \geq \tau_k\right) \leq 1 - \left(1 - \exp\left(T\left(\tau_k\left(\frac{1}{e} - 1\right) + 1\right)\right)\right)^K. \tag{1}$$

For detailed proof, please refer to Appendix A.1. In Theorem 1, we demonstrate the relationship between false recognition and the parameters $T$, $K$, and $\tau_k$ for the traditional FKE method. Our theory shows that as $T$ increases, false recognition can be alleviated, which aligns with the intuition that more tokens provide more accurate information. However, as the capacity $K$ increases, the false positive bound also increases. This implies that with a larger capacity, traditional FKE methods are more likely to mistakenly identify plain text as watermarked text. We can also observe that as $\tau_k$ increases, the bound decreases, indicating that using a stricter threshold can help alleviate false recognition problem.

The key aspect of our proposed DWI method is that we use a subset of tokens to indicate whether the text is watermarked, rather than using all of them to encode key information. To analysize our method, we assume that we use $T' = \lfloor rT \rfloor$ tokens to encode an indicator variable, where $r \in (0, 1)$ represents the ratio of such tokens, and $\lfloor \cdot \rfloor$ denotes the floor function. In Theorem 2, we prove the upper bound for false recognition in the DWI method.

**Theorem 2.** *Consider random variables $u_{ix_i}$ drawn from a uniform distribution on $[0, 1]$. The index $i = [1, \ldots, T']$ represents the $i$th token of the generated text. We calculate the score as $S_d = -\frac{1}{T'} \sum_{i=1}^{T'} \ln(1 - u_{ix_i})$. We regard the sample as watermarked if $S_d \geq \tau_d$, where $\tau_d$ is a*

Figure 1: Numerical comparison of the probability bounds for DWI and FKE methods, presenting the numerical bounds for Theorem 1 and Theorem 2.

| $T$ | $r = 0.2$ | $r = 0.5$ | $r = 0.8$ |
|-----|-----------|-----------|-----------|
| 200 | 9.3 | 3.6 | 1.6 |
| 300 | 21.1 | 6.0 | 2.0 |
| 400 | 48.7 | 10.2 | 2.5 |
| 500 | 114.7 | 17.7 | 3.1 |

Table 1: Lower bounds for $K$ that DWI is better than FKE with $\tau_d = \tau_k = 1.6$ and varying $T$ and $r$.

*threshold parameter. Then the false positive probability is bounded as follows:*

$$\Pr\left(S_d \geq \tau_d\right) \leq \exp\left(T'\left(\tau_d\left(\frac{1}{e} - 1\right) + 1\right)\right). \tag{2}$$

For detailed proof, please see Appendix A.2. Theorem 2 demonstrates the relationship between false recognition and the parameters $T' = \lfloor rT \rfloor$ and $\tau_d$ for the DWI method. Similar to FKE, as $T'$ increases, the false recognition problem can be alleviated. Specifically, when $T$ is fixed, increasing $r$ can help mitigate the false recognition issue. Similarly, we observe that as $\tau_d$ increases, the bound decreases, indicating that using a stricter threshold can further alleviate the false recognition problem. It should be noted that the bound is independent of the capacity $K$, which significantly helps in reducing the false recognition problem especially when $K$ is large.

To provide a detailed comparison between the FKE and DWI methods, we conducted a numerical experiment by plotting the probability bound for Theorem 1 with $K$ ranging from 50 to 500, and the probability bound for Theorem 2 with $r$ ranging from 0.2 to 0.8, as shown in Figure 1. The plot clearly shows that the bound for DWI methods is significantly lower than that of FKE, demonstrating the effectiveness of our approach. Since the number of tokens used in the DWI method is smaller than that used in FKE, we numerically calculated the minimal $K$ value for which DWI's bound outperforms FKE's bound, given specific $T$ and $r$. As shown in Table 1, if $K$ is larger than 114.7, the DWI method's bound is superior for all settings we test. This capacity is relatively small, indicating that, our DWI method outperforms the FKE method for almost any capacity $K$. This conclusion is also supported by our analysis in Lemma 6.

It is a natural extension to combine the FKE and DWI methods to form the HDWI method, as discussed. Here, we also establish a theoretical analysis of the HDWI method to demonstrate its effectiveness. The bound is provided in Theorem 3.

**Theorem 3.** *With the same notation introduced in Theorem 1 and Theorem 2, we use $\lfloor rT \rfloor$ tokens to calculate $S_d$ and $\lfloor (1 - r)T \rfloor$ tokens to calculate $S_k(\xi)$. The hybrid probability $\Pr(S_d > \tau_d \cap \max_\xi S_k(\xi) > \tau_k)$ is bounded as follows:*

$$\Pr\left(S_d \geq \tau_d \cap \max_\xi S_k(\xi) > \tau_k\right) = \Pr\left(S_d \geq \tau_d\right) \cdot \Pr\left(\max_\xi S_k(\xi) > \tau_k\right) \tag{3}$$

$$\leq \exp\left(\lfloor rT \rfloor\left(\tau_d\left(\frac{1}{e} - 1\right) + 1\right)\right)\left(1 - \left(1 - \exp\left(\lfloor (1-r)T \rfloor\left(\tau_k\left(\frac{1}{e} - 1\right) + 1\right)\right)\right)^K\right) \tag{4}$$

Theorem 3 is a straightforward combination of the results from Theorem 1 and Theorem 2. A detailed proof can be found in Appendix A.3. From Theorem 3, a similar conclusion can be observed: as $T$ increases, the bound decreases, indicating an improved ability to alleviate the false recognition problem. Similarly, if $\tau_k$ and $\tau_d$ increase, the bound also decreases, further enhancing performance against false recognition. The effect of $r$ depends on which part is dominant. The bound for the hybrid strategy generally dominates those of method FKE and DWI. It strictly dominates strategy DWI when $r < 1$ and coincides with it when $r = 1$. Likewise, it coincides with strategy FKE when $K = 1$ and strictly dominates it when $K > 1$.

## 5 EXPERIMENTS

### 5.1 EXPERIMENTAL SETUPS

Our experiments following the same setup as Fernandez et al. (2023), using the Guanaco-7b model (Dettmers et al., 2024), an instruction fine-tuned LLaMA model (Touvron et al., 2023a), with the first 1,000 prompts from the Alpaca dataset (Taori et al., 2023). Our dataset consists of 1,000 samples, including a mix of watermarked and unwatermarked text. We utilize a salt key to represent

key IDs ranging from 1 to 1,000. Our framework returns either 'None'—indicating that the text is unwatermarked—or an integer representing the key ID. We use 1000 samples mixed with watermarked and unwatermarked text to test all the methods. To assess the models' capability across a range of proportions of watermarked text, we generated 11 datasets with watermarked text ratios ranging from [0%, 10%, ..., 90%, 100%]. We report the overall scores by averaging the metrics across all datasets with different watermarked ratios. We utilized 500 samples as a development set for hyperparameter selection and another 500 samples as a test set for evaluation with the chosen hyperparameters. To determine the optimal hyperparameters, we employed a grid search on $\tau_d$ and $\tau_k$, exploring values within the range [0.02, 0.04, ..., 7.98, 8.0] for all models. We evaluate our method using three metrics: Accu-I measures the accuracy of determining whether the text is watermarked, irrespective of the correctness of the key prediction. It converts all results to binary outcomes—1 for watermarked and 0 for non-watermarked—and compares these with the gold standard; Accu-O represents the overall accuracy, assessing both the accuracy of watermark indicator predictions and key predictions. and False Positive Ratio (FPR), which indicates the extent of the false recognition problem. We follow Fernandez et al. (2023) in using cosine similarity (Sim) between watermarked and unwatermarked text to evaluate generation quality and information loss. A higher cosine similarity score indicates that the generated watermarked text closely resembles the unwatermarked text, reflecting better quality and minimal information loss. We run all of our models on NVIDIA A100 GPU with 80GB memory and for the inference, we use parallel programming to calcuate the scores for different candidate $\xi$ on a 128 core Intel(R) Xeon(R) Gold 6338 CPU @ 2.00GHz machine.

We compare our DWI method, with the following baseline methods: Full Key Encoding (FKE), Partial Key Encoding (PKE), Hybrid Dual Watermark Inspector (HDWI), Mean Rebalance (MR), and Second Rebalance (SR). Details of these methods have been discussed in Section 3.2. We also compare our models to the MultiBit model (Yoo et al., 2023b; Wang et al., 2023). For the main experiment, we modify the implementation of Yoo et al. (2023b)'s method to align with the FKE framework. For the generalizability study, we use the original implementation from Yoo et al. (2023b); Wang et al. (2023), which involves splitting the vocabulary into colored lists.

## 5.2 MAIN EXPERIMENTS

We compare our proposed DWI and HDWI methods with several baseline models. It can be observed from the results shown in Table 2 that: (1) The HDWI method and its variants outperform all other models in Accu-I and FPR, demonstrating the effectiveness of our proposed methods in detecting watermarked text. The FPR of these models is consistent with our analysis in Theorem 3, further showing the correctness of our theory. (2) Compared to the baseline model FKE, DWI shows a slight decrease in Accu-O. This reduction occurs because only half of the tokens are utilized to encode the key information. However, given that this approach significantly mitigates the false recognition problem, this compromise is considered acceptable. (3) By comparing HDWI, SR, and MR, it shows that SR and MR effectively improve performance, highlighting the effectiveness of these variant strategies to enhance performance. (4) The performance of PKE lags significantly behind FKE. This discrepancy arises because PKE utilizes only half of the tokens to encode the key information and leaves the remaining tokens unused. This comparison highlights the crucial role of the parameter $T$ in influencing performance and further substantiates the validity of our theory. These results also elucidate why HDWI achieves only marginal improvements; using only half of the tokens to encode key information may lead to a decline in performance. (5) Regarding the FPR score, HDWI outperforms both DWI and FKE, further demonstrating the correctness of our analysis as detailed in Section 4.1. (6) The Sim scores across all models are closely clustered around 0.69, suggesting that the quality of the generated text is similar among the models. This also indicates that all models maintain an adequate level of similarity to the unwatermarked text.

## 5.3 INDICATION RATIO $r_d$

In the main experiment, we set $r_d = 50\%$, thereby utilizing half of the tokens to encode whether the text is watermarked, while the remaining tokens encode the key information. In this subsequent experiment, we evaluate the ratios $r_d$ in [10%, 30%, 50%, 70%, 90%] to assess the impact of employing more tokens to encode the watermark indicator. The results, depicted in Figure 2, indicate that (1) employing more tokens for encoding the indicator can substantially mitigate the FPR. (2) It can be observed that as the watermarked text ratio increases, the FPR also rises. This phenomenon occurs because, during threshold tuning on development set, when the ratio of watermarked text approaches zero, the model tends to select a threshold that directly classifies all samples as unwa-

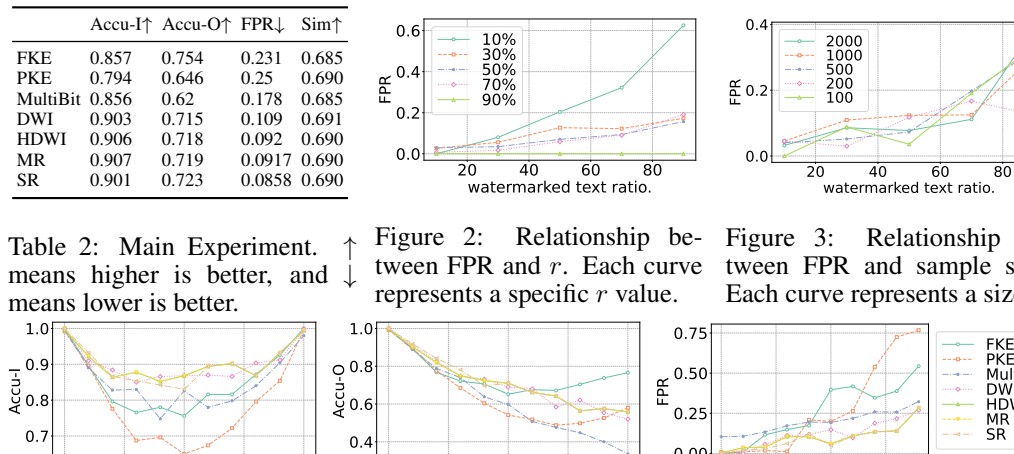

| | Accu-I↑ | Accu-O↑ | FPR↓ | Sim↑ |
|---|---|---|---|---|
| FKE | 0.857 | 0.754 | 0.231 | 0.685 |
| PKE | 0.794 | 0.646 | 0.25 | 0.690 |
| MultiBit | 0.856 | 0.62 | 0.178 | 0.685 |
| DWI | 0.903 | 0.715 | 0.109 | 0.691 |
| HDWI | 0.906 | 0.718 | 0.092 | 0.690 |
| MR | 0.907 | 0.719 | 0.0917 | 0.690 |
| SR | 0.901 | 0.723 | 0.0858 | 0.690 |

Table 2: Main Experiment. ↑ means higher is better, and ↓ means lower is better.

Figure 2: Relationship between FPR and $r$. Each curve represents a specific $r$ value.

Figure 3: Relationship between FPR and sample size. Each curve represents a size.

Figure 4: Watermarked text ratio results. The figures illustrate the relationship between the watermarked text ratio $r$ and the corresponding metrics. Each plot represents a specific metric, with metrics calculated by varying the thresholds $\tau_k$ and $\tau_d$ according to the watermark ratio $r$.

termarked and thus make less false positive error. When the watermarked text ratio is high, the threshold is adjusted to classify more samples as watermarked. This setting leads to an increase in false positive errors.

## 5.4 SAMPLE SIZE EXPERIMENT

To demonstrate that our experiments used a reasonable sample size and that our results are not sensitive to sample size, we performed an experiment by varying the total sample size within the range of $[100, 200, 500, 1000, 2000]$. As observed in Figure 3, the results did not differ significantly with changes in sample size. This indicates that the sample size we chose is suitable for our current experiment and that our proposed method has good generalizability, not relying heavily on the number of samples. It can be observed that as the ratio of watermarked text increases, the FPR also rises. The underlying reason for this phenomenon is the same as discussed in Section 5.3.

## 5.5 WATERMARKED TEXT RATIO

The thresholds $\tau_k$ and $\tau_d$ are critical in determining whether a text is watermarked, with their optimal values varying according to the watermarked text ratio. The metrics presented in the main experiment are averaged across datasets with differing ratios of watermarked text. To provide a comprehensive analysis of performance across various watermarked text ratios, we examine the relationship between the watermarked text ratio and the metrics Accu-I, Accu-O, and FPR. As illustrated in Figure 4, the following observations are made: (1) As the watermarked text ratio increases from 0% to 100%, Accu-I initially decreases and then rises after the 50% mark. This trend occurs because, when the watermark ratio is 0%, tuning the threshold on the development set to a very high value results in classifying all samples as unwatermarked, thereby leading to optimal performance. A similar situation arises when the watermark ratio nears 100%, tuning the threshold to classify all samples as watermarked yields the best performance on both the development set and the test set. (2) Accu-O decreases as the watermark ratio increases, due to models "overfitting" to predict all samples as unwatermarked when the ratio is 0%. As the ratio increases, predicting the exact key ID becomes more challenging than merely predicting whether the text is watermarked. However, Accu-O slightly increases as the watermark ratio approaches 100% for FKE and PKE. This improvement occurs because the models are tuned to avoid predicting any samples as unwatermarked. (3) With an increasing watermarked text ratio, FPR also increases. This is because, at low watermark ratios, the tuned thresholds are set very high, making it unlikely for any sample to be classified as watermarked, thus eliminating false positive inferences. (4) It is evident that our proposed DWI and HDWI models outperform the FKE model across nearly all watermarked text ratios, demonstrating the effectiveness of our approach in various scenarios.

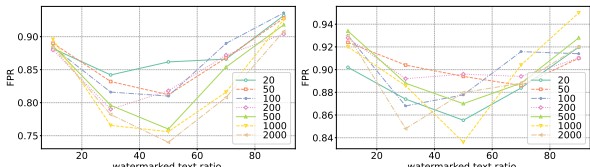 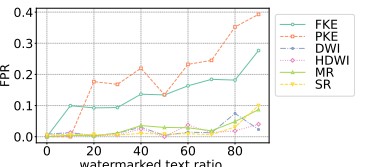

Figure 5: Key capacity results for FKE (left) and HDWI (right). The figures illustrate the relationship between the watermarked text ratio and FPR for varying key capacities ( $K$ ranges from 20 to 2000).

Figure 6: Generalizability. Based on Kirchenbauer et al. (2023a)'s methods, showing the relationship between FPR and watermarked text ratio ($r$).

| | Accu-I↑ | Accu-O↑ | FPR↓ | | Accu-I↑ | Accu-O↑ | FPR↓ |
|---|---|---|---|---|---|---|---|
| 20 | 0.877 | 0.824 | 0.178 | 20 | 0.887 | 0.786 | 0.173 |
| 50 | 0.866 | 0.796 | 0.234 | 50 | 0.904 | 0.786 | 0.149 |
| 100 | 0.867 | 0.799 | 0.233 | 100 | 0.901 | 0.777 | 0.12 |
| 200 | 0.853 | 0.777 | 0.260 | 200 | 0.904 | 0.757 | 0.127 |
| 500 | 0.843 | 0.748 | 0.258 | 500 | 0.902 | 0.742 | 0.103 |
| 1000 | 0.832 | 0.735 | 0.289 | 1000 | 0.899 | 0.717 | 0.141 |
| 2000 | 0.824 | 0.721 | 0.295 | 2000 | 0.893 | 0.689 | 0.095 |

| | Accu-I↑ | Accu-O↑ | FPR↓ | Sim↑ |
|---|---|---|---|---|
| FKE | 0.926 | 0.883 | 0.120 | 0.550 |
| PKE | 0.898 | 0.835 | 0.175 | 0.562 |
| MultiBit | 0.936 | 0.863 | 0.0412 | 0.561 |
| DWI | 0.959 | 0.910 | 0.0178 | 0.562 |
| HDWI | 0.955 | 0.909 | 0.0142 | 0.562 |
| MR | 0.957 | 0.903 | 0.0243 | 0.556 |
| SR | 0.948 | 0.910 | 0.0174 | 0.562 |

Table 3: Key capacity results for FKE (left) and HDWI (right) methods at varying key capacities (20 to 2000). FKE performance decreases as K increases, while HDWI remains stable across different key capacities.

Table 4: Generalizability. Based on Kirchenbauer et al. (2023a)'s methods, which divide the dictionary into green and red sets.

### 5.6 KEY CAPACITY EXPERIMENT

To demonstrate the model's capacities for different key capacities $K$, we compare the FKE and HDWI methods with the capacity $K$ ranging in $[20, 50, 100, 200, 500, 1000, 2000]$. We present the average scores for FKE and HDWI in Section 5.6 (left) and Section 5.6 (right), respectively. Additionally, we provide a detailed breakdown of scores relative to different watermarked text ratios in Figure 5, respectively. It can be observed from the results that: (1) In the FKE model, since it relies entirely on the maximal score of all keys in the key space, the performance decreases significantly as $K$ increases. This is supported and guaranteed by our analysis in Theorem 1. (2) The HDWI results show no significant differences for different key capacities in our HDWI method. This is because the indicator is the dominant part and can ensure the FPR avoids the influence of the total key count.

### 5.7 GENERALIZABILITY STUDY

Our previous experiments and analysis were based on the watermarking methods proposed by Aaronson & Kirchner (2023); Fernandez et al. (2023). To demonstrate that our approach is generalizable to other methods, we have adapted our technique to models introduced by Kirchenbauer et al. (2023a); Fernandez et al. (2023), which categorize the word dictionary into green and red lists based on the keys. We report the scores for different models in Table 4 and provide a detailed breakdown of scores relative to varying watermarked text ratios in Figure 6. The results indicate that (1) across different models, our models consistently outperform the baseline FKE model significantly. (2) Our proposed method significantly mitigates the false recognition issue across all watermarked text ratios. These observations demonstrate that our proposed method can be generalized to other watermarking methods and effectively alleviate the false recognition problem.

## 6 CONCLUSION

In this paper, we address the false recognition problem in watermarking methods for text generated by LLMs. We establish a rigorous theoretical bound demonstrating the inherent inevitability of false positive errors in Information Conveyance watermarking techniques like FKE. To mitigate this problem, we introduce a novel DWI method that jointly encodes indicator and key information. Furthermore, we present a analysis of our proposed method and validate it through extensive empirical experiments. Our results, both theoretical and empirical, indicate that the DWI method and its variants effectively reduces the false positive ratio, thereby alleviating the false recognition problem. This enhancement in watermarking reliability can significantly promote the trustworthiness of LLM-generated content.

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

# Appendix. Supplementary Material

## A.1  PROOF OF THEOREM 1

**Theorem 1.** *Consider random variables $u_{ix_i}(\xi)$ drawn from a uniform distribution over $[0,1]$, where $\xi = [1, \ldots, K]$ represents the key, and $K$ denotes the total key capacity. The index $i = [1, \ldots, T]$ refers to the ith token in the generated sequence. Each $u_i$ is a vector in $\mathbb{R}^V$, where $V$ is the vocabulary size, and $u_{ix_i}$ corresponds to the $x_i$th token in $u_i$. The score is calculated as: $S_k(\xi) = -\frac{1}{T} \sum_{i=1}^{T} \ln(1 - u_{ix_i}(\xi))$. We consider the sample is watermarked if $\max_\xi S_k(\xi) \geq \tau_k$, where $\tau_k$ is a threshold parameter. Then the false positive probability is bounded as follows:*

$$\Pr\left(\max_\xi S_k(\xi) \geq \tau_k\right) \leq 1 - \left(1 - \exp\left(T\left(\tau_k\left(\frac{1}{e} - 1\right) + 1\right)\right)\right)^K. \tag{1}$$

*Proof Sketch.* From Lemma 1, we know that if $r$ is uniformly distributed over $[0,1]$, then $X = -\ln(1 - r)$ follows an exponential distribution with parameter 1. According to Lemma 2, if $u_{ix_i}(\xi)$ are independent and uniformly distributed over $[0,1]$, then $S_k(\xi) = -\frac{1}{T} \sum_{i=1}^{T} \ln(1 - u_{ix_i}(\xi)) \sim$ Gamma$(T, \frac{1}{T})$. Using Lemma 3, given $X \sim$ Gamma$(T, \frac{1}{T})$, with probability $1 - \delta$, $X \leq \frac{\frac{\log \delta}{T} - 1}{1/e - 1}$. Therefore, for $S_k(\xi) \sim$ Gamma$(T, \frac{1}{T})$, with probability $1 - \delta$, $S_k(\xi) \leq \frac{\frac{\log \delta}{T} - 1}{1/e - 1}$. Given this bound for each $S_k(\xi)$, we use Lemma 4 to bound the probability of the maximum $S_k(\xi)$ over $\xi$. Specifically, Lemma 4 states that $\Pr\left(\max_\xi S_k(\xi) \leq \tau_k\right) \geq \left(1 - \exp\left(T\left(\tau_k\left(\frac{1}{e} - 1\right) + 1\right)\right)\right)^K$. Taking the complement, we get $\Pr\left(\max_\xi S_k(\xi) \geq \tau_k\right) \leq 1 - \left(1 - \exp\left(T\left(\tau_k\left(\frac{1}{e} - 1\right) + 1\right)\right)\right)^K$. This completes the proof of the theorem. $\square$

*Proof.* We will prove the theorem step-by-step using the provided lemmas.

**Step 1: Showing the Transformation is Exponential**

From Lemma 1, we know that if $r$ is uniformly distributed over $[0,1]$, then $X = -\ln(1-r)$ follows an exponential distribution with parameter 1, i.e., $X \sim \text{Exp}(1)$.

**Step 2: Distribution of $S_k(\xi)$**

From Lemma 2, we know that if $u_{ix_i}(\xi)$ are independent and uniformly distributed over $[0,1]$, then

$$S_k(\xi) = -\frac{1}{T} \sum_{i=1}^{T} \ln(1 - u_{ix_i}(\xi)) \sim \text{Gamma}\left(T, \frac{1}{T}\right).$$

**Step 3: Bounding $S_k(\xi)$**

Using Lemma 3, given $X \sim$ Gamma$(T, \frac{1}{T})$, with probability $1 - \delta$,

$$X \leq \frac{\frac{\log \delta}{T} - 1}{1/e - 1}.$$

Therefore, for $S_k(\xi) \sim$ Gamma$(T, \frac{1}{T})$, with probability $1 - \delta$,

$$S_k(\xi) \leq \frac{\frac{\log \delta}{T} - 1}{1/e - 1}.$$

**Step 4: Probability Bound on Maximum $S_k(\xi)$**

Given $S_k(\xi) \leq \frac{\frac{\log \delta}{T} - 1}{1/e - 1}$ with probability $1 - \delta$ for each $i$, we use Lemma 4 to bound the probability of the maximum $S_k(\xi)$.

From Lemma 4, we have:

$$\Pr\left(\max_\xi S_k(\xi) \leq \tau_k\right) \geq \left(1 - \exp\left(T\left(\tau_k\left(\frac{1}{e} - 1\right) + 1\right)\right)\right)^K.$$

**Step 5: Complement of Maximum Bound**

To find the probability that the maximum of $S_k(\xi)$ exceeds $\tau_k$, we take the complement of the bound derived above:

$$\Pr\left(\max_{\xi} S_k(\xi) \geq \tau_k\right) \leq 1 - \left(1 - \exp\left(T\left(\tau_k\left(\frac{1}{e} - 1\right) + 1\right)\right)\right)^K.$$

**Conclusion**

We have shown that the probability of the maximum score $\max_{\xi} S_k(\xi)$ being greater than or equal to $\tau_k$ is bounded by

$$\Pr\left(\max_{\xi} S_k(\xi) \geq \tau_k\right) \leq 1 - \left(1 - \exp\left(T\left(\tau_k\left(\frac{1}{e} - 1\right) + 1\right)\right)\right)^K.$$

This completes the proof of the theorem. With probability $1 - \delta$,

$$\Pr\left(\max_{\xi} S_k(\xi) \geq \tau_k\right) \leq 1 - \left(1 - \exp\left(T\left(\tau_k\left(\frac{1}{e} - 1\right) + 1\right)\right)\right)^K.$$

$\square$

## A.2   PROOF OF THEOREM 2

**Theorem 2.** *Consider random variables $u_{ix_i}$ drawn from a uniform distribution on $[0,1]$. The index $i = [1, \ldots, T']$ represents the $i$th token of the generated text. We calculate the score as $S_d = -\frac{1}{T'}\sum_{i=1}^{T'}\ln(1 - u_{ix_i})$. We regard the sample as watermarked if $S_d \geq \tau_d$, where $\tau_d$ is a threshold parameter. Then the false positive probability is bounded as follows:*

$$\Pr\left(S_d \geq \tau_d\right) \leq \exp\left(T'\left(\tau_d\left(\frac{1}{e} - 1\right) + 1\right)\right). \tag{2}$$

*Proof Sketch.* Given $u_{ix_i}$ drawn from a uniform distribution on $[0,1]$, we transform $u_{ix_i}$ using $-\ln(1 - u_{ix_i})$, which follows an exponential distribution with parameter 1, as shown in Lemma 1. The score $S_d$ is then the average of $T'$ such transformed variables, scaled by $-\frac{1}{T'}$, which, by Lemma 2, follows a Gamma distribution with shape parameter $T'$ and scale parameter $\frac{1}{T'}$. Using the bound from Lemma 3, with probability $1 - \delta$, $S_d$ is less than or equal to a certain function of $\log \delta$. By expressing $\delta$ in terms of $\tau_d$ and solving, we derive that the false positive probability $\Pr(S_d \geq \tau_d)$ is bounded by an exponential function $\exp\left(T'\left(\tau_d\left(\frac{1}{e} - 1\right) + 1\right)\right)$. Thus, the false positive probability is bounded as claimed in the theorem. $\square$

*Proof.* We will prove this theorem using the following lemmas.

From Lemma 1, we know that if $u_{ix_i} \sim \text{Uniform}(0, 1)$, then $-\ln(1 - u_{ix_i}) \sim \text{Exp}(1)$.

Using Lemma 2, the sum of $T'$ independent exponential random variables follows a Gamma distribution with shape parameter $T'$ and scale parameter $\frac{1}{T'}$:

$$S_d = -\frac{1}{T'}\sum_{i=1}^{T'}\ln(1 - u_{ix_i}) \sim \text{Gamma}(T', \frac{1}{T'}).$$

From Lemma 3, we know that for $X \sim \text{Gamma}(T', \frac{1}{T'})$, with shape parameter $T'$ and scale parameter $\frac{1}{T'}$, with probability $1 - \delta$:

$$X \leq \frac{\frac{\log \delta}{T'} - 1}{1/e - 1}.$$

Adapting this for our case where the shape parameter is $T'$, with probability $1 - \delta$:

$$S_d \leq \frac{\frac{\log \delta}{T'} - 1}{1/e - 1}.$$

Let's express $\delta$ as a function of $\tau_d$. We set:

$$\tau_d = \frac{\frac{\log \delta}{T'} - 1}{1/e - 1}.$$

Solving for $\log \delta$:

$$\tau_d(1/e - 1) = \frac{\log \delta}{T'} - 1,$$

$$\log \delta = T'(\tau_d(1/e - 1) + 1).$$

Thus, we have:

$$\delta = \exp\left(T'(\tau_d(1/e - 1) + 1)\right).$$

The false positive probability $\Pr(S_d \geq \tau_k)$ is given by $\delta$:

$$\Pr(S_d \geq \tau_k) = \exp\left(T'(\tau_d(1/e - 1) + 1)\right).$$

Thus, we have shown that the false positive probability is bounded as follows: with probability $1 - \delta$,

$$\Pr\left(S_d \geq \tau_k\right) \leq \exp\left(T'\left(\tau_d\left(\frac{1}{e} - 1\right) + 1\right)\right).$$

$\square$

## A.3 PROOF OF THEOREM 3

**Theorem 3.** *With the same notation introduced in Theorem 1 and Theorem 2, we use $\lfloor rT \rfloor$ tokens to calculate $S_d$ and $\lfloor (1-r)T \rfloor$ tokens to calculate $S_k(\xi)$. The hybrid probability $\Pr(S_d > \tau_d \cap \max_\xi S_k(\xi) > \tau_k)$ is bounded as follows:*

$$\Pr\left(S_d \geq \tau_d \cap \max_\xi S_k(\xi) > \tau_k\right) = \Pr\left(S_d \geq \tau_d\right) \cdot \Pr\left(\max_\xi S_k(\xi) > \tau_k\right) \tag{3}$$

$$\leq \exp\left(\lfloor rT \rfloor \left(\tau_d\left(\frac{1}{e} - 1\right) + 1\right)\right)\left(1 - \left(1 - \exp\left(\lfloor (1-r)T \rfloor \left(\tau_k\left(\frac{1}{e} - 1\right) + 1\right)\right)\right)^K\right) \tag{4}$$

*Proof.* We start by considering the two probabilities involved in the hybrid probability $\Pr(S_d \geq \tau_d \cap \max_\xi S_k(\xi) > \tau_k)$. By the definition of joint probability for independent events, we can express the hybrid probability as the product of the individual probabilities:

$$\Pr\left(S_d \geq \tau_d \cap \max_\xi S_k(\xi) > \tau_k\right) = \Pr\left(S_d \geq \tau_d\right) \cdot \Pr\left(\max_\xi S_k(\xi) > \tau_k\right).$$

Given that $S_d$ is calculated using $\lfloor rT \rfloor$ tokens and $S_k(\xi)$ is calculated using $\lfloor (1-r)T \rfloor$ tokens, we can apply the bounds from Theorem 2 and Theorem 1 respectively.

First, by applying the bound from Theorem 2 to the probability $\Pr(S_d \geq \tau_d)$, we have:

$$\Pr\left(S_d \geq \tau_d\right) \leq \exp\left(\lfloor rT \rfloor \left(\tau_d\left(\frac{1}{e} - 1\right) + 1\right)\right).$$

Next, by applying the bound from Theorem 1 to the probability $\Pr(\max_\xi S_k(\xi) > \tau_k)$, we obtain:

$$\Pr\left(\max_\xi S_k(\xi) > \tau_k\right) \leq 1 - \left(1 - \exp\left(\lfloor (1-r)T \rfloor \left(\tau_k\left(\frac{1}{e} - 1\right) + 1\right)\right)\right)^K.$$

Thus, combining these two results, the hybrid probability can be bounded as follows:

$$\Pr\left(S_d \geq \tau_d \cap \max_{\xi} S_k(\xi) > \tau_k\right) \leq \exp\left(\lfloor rT \rfloor \left(\tau_d\left(\frac{1}{e} - 1\right) + 1\right)\right) \cdot \tag{5}$$

$$\left(1 - \left(1 - \exp\left(\lfloor (1-r)T \rfloor \left(\tau_k\left(\frac{1}{e} - 1\right) + 1\right)\right)\right)^K\right). \tag{6}$$

This completes the proof. □

### A.4 PROOF OF THE EQUIVALENCE OF GUMBEL-MAX TRICK

**Proposition 1.** *Consider a discrete distribution $p = (p_1, \ldots, p_V)$ and $V$ random variables $u = (u_1, \ldots, u_V)$ such that $u_v$ are i.i.d. with $u_v \sim \mathcal{U}_{[0,1]}$. Let $V^\star = \arg\max_v u_v^{1/p_v}$. Define $G_v = \log(p_v) + g_v$, where $g_v = -\log(-\log(u_v))$. Then*

$$V^\star = G^\star$$

*Proof.*
$$\begin{aligned}
\arg\max_v G_v &= \arg\max_v \left(\log(p_v) + g_v\right) \\
&= \arg\max_v \left(\log(p_v) - \log(-\log(u_v))\right) \\
&= \arg\max_v \exp\left(\log(p_v) - \log(-\log(u_v))\right) \\
&= \arg\max_v \left(\exp(\log(p_v)) \cdot \exp(-\log(-\log(u_v)))\right) \\
&= \arg\max_v \left(p_v \cdot \frac{1}{-\log(u_v)}\right) \\
&= \arg\min_v \left(-\frac{\log(u_v)}{p_v}\right) \\
&= \arg\max_v \left(\frac{\log(u_v)}{p_v}\right) \\
&= \arg\max_v \left(\log(u_v^{1/p_v})\right) \\
&= \arg\max_v \left(u_v^{1/p_v}\right)
\end{aligned}$$

Therefore,
$$V^\star = \arg\max_v u_v^{1/p_v}$$

Thus, the theorem is proved:
$$V^\star = G^\star$$

□

### A.5 LEMMAS

**Lemma 1.** *Let $r$ be a random variable uniformly distributed over the interval $[0,1]$. Define $X = -\ln(1-r)$. Then $X$ follows an exponential distribution with parameter 1, i.e., $X \sim Exp(1)$.*

*Proof.* To show that $X = -\ln(1-r)$ follows an exponential distribution with parameter 1, we first find the cumulative distribution function (CDF) of $X$.

For any $x \geq 0$,

$$
\begin{aligned}
F_X(x) &= P(X \leq x) \\
&= P(-\ln(1-r) \leq x) \\
&= P(\ln(1-r) \geq -x) \\
&= P(1 - r \geq e^{-x}) \\
&= P(r \leq 1 - e^{-x}).
\end{aligned}
$$

Since $r$ is uniformly distributed over $[0,1]$, its CDF is $F_r(r) = r$. Therefore,

$$
F_X(x) = 1 - e^{-x}, \quad \text{for } x \geq 0.
$$

Next, we differentiate the CDF to obtain the probability density function (PDF):

$$
f_X(x) = \frac{d}{dx} F_X(x) = \frac{d}{dx}(1 - e^{-x}) = e^{-x}, \quad \text{for } x \geq 0.
$$

The PDF $f_X(x) = e^{-x}$ is the PDF of an exponential distribution with parameter 1. Therefore, $X \sim \text{Exp}(1)$. □

**Lemma 2.** *Let $r_i$ be independent and uniformly distributed over the interval $[0,1]$ for $i = 1, 2, \ldots, T$. Define $S = -\frac{1}{T} \sum_{i=1}^{T} \ln(1 - r_i)$. Then $S$ follows a Gamma distribution with shape parameter $T$ and scale parameter $\frac{1}{T}$, i.e., $S \sim \text{Gamma}(T, \frac{1}{T})$.*

*Proof.* From Lemma 1, we know that if $r$ is uniformly distributed over $[0,1]$, then $X = -\ln(1 - r) \sim \text{Exp}(1)$.

Given that $r_i \sim \text{Uniform}(0,1)$, it follows that $u_i = -\ln(1 - r_i) \sim \text{Exp}(1)$ for each $i$.

Now, consider the sum of $T$ such independent exponential random variables:

$$
Y = \sum_{i=1}^{T} u_i
$$

Since the sum of $T$ independent $\text{Exp}(1)$ random variables follows a Gamma distribution with shape parameter $T$ and scale parameter 1, we have:

$$
Y \sim \text{Gamma}(T, 1)
$$

Next, consider the scaled variable:

$$
S = \frac{Y}{T}
$$

Since $Y \sim \text{Gamma}(T, 1)$, scaling $Y$ by $1/T$ (which is equivalent to dividing by $T$) gives us a new Gamma distributed random variable with the same shape parameter $T$ and a scale parameter of $1/T$. Therefore:

$$
S \sim \text{Gamma}\left(T, \frac{1}{T}\right)
$$

Thus, we have shown that $S = -\frac{1}{T} \sum_{i=1}^{T} \ln(1 - r_i)$ follows a Gamma distribution with shape parameter $T$ and scale parameter $\frac{1}{T}$. □

**Lemma 3.** *Given $X \sim \text{Gamma}(T, \frac{1}{T})$, with shape parameter $T$ and scale parameter $\frac{1}{T}$, we can state:*

*With probability $1 - \delta$,*

$$
X \leq \frac{\frac{\log \delta}{T} - 1}{1/e - 1}.
$$

*Proof.* We use the Chernoff bound to derive this result.

First, recall the moment generating function (MGF) of $X \sim \text{Gamma}(T, \frac{1}{T})$:

$$M_X(t) = \mathbb{E}[e^{tX}] = \left(1 - \frac{t}{T}\right)^{-T},$$

for $t < T$.

Using the Chernoff bound, for any $t > 0$, we have:

$$\mathbb{P}(X \geq a) = \mathbb{P}(e^{tX} \geq e^{ta}) \leq \frac{\mathbb{E}[e^{tX}]}{e^{ta}} = \frac{M_X(t)}{e^{ta}}.$$

Substituting the MGF, we get:

$$\mathbb{P}(X \geq a) \leq \frac{\left(1 - \frac{t}{T}\right)^{-T}}{e^{ta}}.$$

To optimize this bound, we need to minimize the right-hand side with respect to $t$. Therefore, we have:

$$\log\left(\frac{\left(1 - \frac{t}{T}\right)^{-T}}{e^{ta}}\right) = -T \log\left(1 - \frac{t}{T}\right) - ta.$$

Differentiate with respect to $t$ and set the derivative to zero to find the optimal $t$:

$$\frac{d}{dt}\left(-T \log\left(1 - \frac{t}{T}\right) - ta\right) = 0$$

$$-T \cdot \left(-\frac{1}{T} \cdot \frac{1}{1 - \frac{t}{T}}\right) - a = 0$$

$$\frac{1}{1 - \frac{t}{T}} - a = 0$$

$$1 - \frac{1}{a} = \frac{t}{T}$$

$$t = T\left(1 - \frac{1}{a}\right).$$

Substituting $t = T\left(1 - \frac{1}{a}\right)$ back into the Chernoff bound, we have:

$$\mathbb{P}(X \geq a) \leq \exp\left(-T \log\left(1 - \left(1 - \frac{1}{a}\right)\right) - T\left(1 - \frac{1}{a}\right)a\right).$$

Simplifying further:

$$\mathbb{P}(X \geq a) \leq \exp\left(-T \log\left(\frac{1}{a}\right) - T(a - 1)\right).$$

For $a > 0$, we can simplify the expression:

$$\mathbb{P}(X \geq a) \leq \exp\left(T \log(a) - T(a - 1)\right)$$

$$\leq \exp\left(T\frac{a}{e} - T(a - 1)\right)$$

$$= \exp\left(T(\frac{a}{e} - a + 1)\right)$$

Setting this bound to $\delta$, we get:

$$\exp\left(T(\frac{a}{e} - a + 1)\right) = \delta.$$

Taking the natural logarithm:

$$T(\frac{a}{e} - a + 1) = \log\delta,$$

$$a = \frac{\frac{\log\delta}{T} - 1}{1/e - 1},$$

Therefore, with probability $1 - \delta$:

$$X \leq \frac{\frac{\log\delta}{T} - 1}{1/e - 1}.$$

$\square$

**Lemma 4.** *Given random variables $u_1, u_2, \ldots, u_K$ where $K > 0$, such that with probability $1 - \delta$:*

$$u_i \leq \frac{\frac{\log\delta}{T} - 1}{1/e - 1},$$

*it follows that:*

$$\Pr\left(\max_i u_i \leq s\right) \geq \left(1 - \exp\left(T\left(s\left(\frac{1}{e} - 1\right) + 1\right)\right)\right)^K.$$

*Proof.* Given the condition:

$$\Pr\left(u_i \leq \frac{\frac{\log\delta}{T} - 1}{1/e - 1}\right) \geq 1 - \delta,$$

we denote:

$$b = \frac{\frac{\log\delta}{T} - 1}{1/e - 1}.$$

We aim to express this condition in terms of $s$ and derive a bound for:

$$\Pr\left(\max_i u_i \leq s\right).$$

First, consider:

$$\Pr\left(u_i \leq b\right) \geq 1 - \delta.$$

We need to find a function of $s$ that relates $\delta$ to $s$. Suppose $s \geq b$. Then:

$$\Pr\left(u_i \leq s\right) \geq \Pr\left(u_i \leq b\right) \geq 1 - \delta.$$

We aim to find the probability that all $u_i$ are less than or equal to $s$:

$$\Pr\left(\max_i u_i \leq s\right) = \Pr\left(u_1 \leq s, u_2 \leq s, \ldots, u_K \leq s\right).$$

Assuming the $u_i$ are independent, we can write:

$$\Pr\left(u_1 \leq s, u_2 \leq s, \ldots, u_K \leq s\right) = \prod_{i=1}^K \Pr\left(u_i \leq s\right).$$

Since:
$$\Pr\left(u_i \le s\right) \ge 1 - \delta,$$

we have:
$$\Pr\left(\max_i u_i \le s\right) \ge (1 - \delta)^K.$$

Now, we need to express $\delta$ in terms of $s$. Recall the expression for $b$:
$$b = \frac{\frac{\log \delta}{T} - 1}{1/e - 1}.$$

Solving for $\log \delta$, we get:
$$b(1/e - 1) = \frac{\log \delta}{T} - 1,$$
$$b(1/e - 1) + 1 = \frac{\log \delta}{T},$$
$$T\left(b(1/e - 1) + 1\right) = \log \delta,$$
$$\delta = \exp\left(T\left(b(1/e - 1) + 1\right)\right).$$

Now, substitute $b = s$:
$$\delta = \exp\left(T\left(s\left(\frac{1}{e} - 1\right) + 1\right)\right).$$

Hence:
$$\Pr\left(\max_i u_i \le s\right) \ge \left(1 - \exp\left(T\left(s\left(\frac{1}{e} - 1\right) + 1\right)\right)\right)^K.$$

This completes the proof of the lemma. $\square$

**Lemma 5.** *Given a random variable $X$, such that with probability $1 - \delta$:*
$$X \le \frac{\frac{\log \delta}{rT} - 1}{1/e - 1},$$

*it follows that:*
$$\Pr\left(X \le s\right) \ge 1 - \exp\left(rT\left(s\left(\frac{1}{e} - 1\right) + 1\right)\right).$$

*Proof.* Given the condition:
$$\Pr\left(X \le \frac{\frac{\log \delta}{rT} - 1}{1/e - 1}\right) \ge 1 - \delta,$$

we denote:
$$b = \frac{\frac{\log \delta}{rT} - 1}{1/e - 1}.$$

We aim to express $\delta$ as a function of $s$ and find the probability bound for $X \le s$.

Rearranging the expression for $b$:
$$b = \frac{\frac{\log \delta}{rT} - 1}{1/e - 1},$$

we solve for $\log \delta$:
$$b\left(\frac{1}{e} - 1\right) = \frac{\log \delta}{rT} - 1,$$
$$b\left(\frac{1}{e} - 1\right) + 1 = \frac{\log \delta}{rT},$$
$$rT\left(b\left(\frac{1}{e} - 1\right) + 1\right) = \log \delta,$$

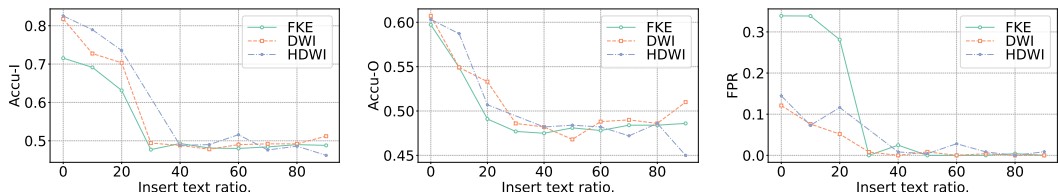

Figure 7: Insertion attack results. The figure shows the impact of varying insertion ratios (10% to 90%) on the metrics Accu-I, Accu-O, and FPR for different watermarking methods (FKE, DWI, HDWI).

$$\delta = \exp\left(rT\left(b\left(\frac{1}{e}-1\right)+1\right)\right).$$

Next, we relate $b$ to $s$. Suppose $s \geq b$, then:

$$\Pr\left(X \leq s\right) \geq \Pr\left(X \leq b\right) \geq 1-\delta.$$

Substitute $b$ with $s$:

$$b = s.$$

Now we have:

$$\delta = \exp\left(rT\left(s\left(\frac{1}{e}-1\right)+1\right)\right).$$

Thus:

$$\Pr\left(X \leq s\right) \geq 1-\delta,$$

where $\delta = \exp\left(rT\left(s\left(\frac{1}{e}-1\right)+1\right)\right)$.

Therefore:

$$\Pr\left(X \leq s\right) \geq 1-\exp\left(rT\left(s\left(\frac{1}{e}-1\right)+1\right)\right).$$

This completes the proof of the lemma. $\qquad\square$

**Lemma 6.** *When* $K \geq \frac{\ln(1-\exp(rT(s(\frac{1}{e}-1)+1)))}{\ln(1-\exp(T(s(\frac{1}{e}-1)+1)))}$, *it follows that:*

$$1-\exp\left(rT\left(s\left(\frac{1}{e}-1\right)+1\right)\right) \geq \left(1-\exp\left(T\left(s\left(\frac{1}{e}-1\right)+1\right)\right)\right)^{K}.$$

*Proof.*

$$1-\exp\left(rT\left(s\left(\frac{1}{e}-1\right)+1\right)\right) \geq \left(1-\exp\left(T\left(s\left(\frac{1}{e}-1\right)+1\right)\right)\right)^{K}$$

$$\ln\left(1-\exp\left(rT\left(s\left(\frac{1}{e}-1\right)+1\right)\right)\right) \geq K\ln\left(1-\exp\left(T\left(s\left(\frac{1}{e}-1\right)+1\right)\right)\right)$$

$$K \geq \frac{\ln\left(1-\exp\left(rT\left(s\left(\frac{1}{e}-1\right)+1\right)\right)\right)}{\ln\left(1-\exp\left(T\left(s\left(\frac{1}{e}-1\right)+1\right)\right)\right)}$$

This completes the proof.

$\qquad\square$

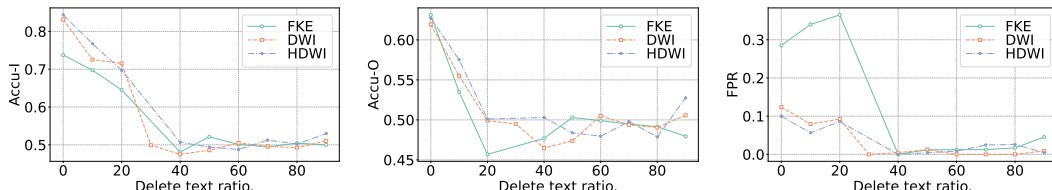

Figure 8: Deletion attack results. The figure illustrates the effect of varying deletion ratios (10% to 90%) on the metrics Accu-I, Accu-O, and FPR for different watermarking methods (FKE, DWI, HDWI).

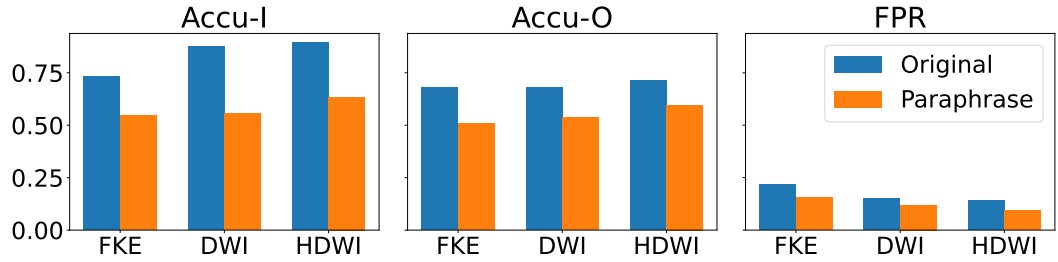

Figure 9: Paraphrase attack results. The figure compares the performance of watermarking methods (FKE, DWI, HDWI) on original and paraphrased text, showing metrics Accu-I, Accu-O, and FPR for a watermarked ratio $r = 0.5$ .

### A.6 INSERTION AND DELETION ATTACK

Following Fernandez et al. (2023), we also perform insertion and deletion attacks, randomly inserting or deleting tokens from the generated text to assess whether such modifications can effectively remove the watermark. We vary the insertion/deletion ratios in the range $[10\%, \cdots, 90\%]$. For instance, if the insertion ratio is 10%, this indicates that we insert tokens amounting to 10% of the total sequence length. Similarly, a deletion ratio of 10% means removing 10% of the tokens from the generated sequence. The experimental results are presented in Figure 7 and Figure 8 respectively. The results indicate the following observations: (1) As the insertion/deletion ratio increases, all scores decrease. This is expected, as modifying more tokens introduces additional noise, making it increasingly difficult to classify the tokens. (2) Our proposed DWI and HDWI methods perform nearly identically to the original FKE method, demonstrating that our approach retains the same robustness capabilities as the original methods.

### A.7 PARAPHRASE ATTACK

We conduct a paraphrase attack to evaluate the robustness of the proposed methods. We set a watermarked ratio $r = 0.5$ to test whether the models can differentiate watermarked text. We use Parrot_Paraphraser[2], a toolkit designed to rephrase sentences generated with watermarks, and we use the same detection tool to detect the watermark and key information. The results are shown in Figure 9. We can observe that (1) our proposed DWI and HDWI models outperform the FKE method, (2) although accuracy decreases after the paraphrase attack, it remains above 0.5, indicating that the methods can still recognize watermarked text and associated keys, and (3) the FPR decreases after the attack because the models are more likely to classify text as unwatermarked. This outcome is expected because, after the paraphrase attack, some previously watermarked text can no longer be detected.

---

[2] https://github.com/PrithivirajDamodaran/Parrot_Paraphraser

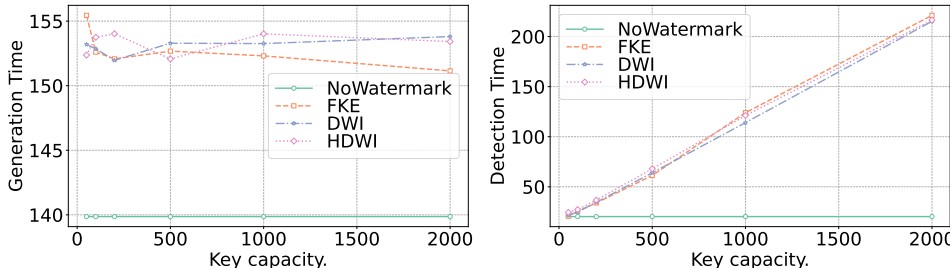

Figure 10: Runtime analysis. The left plot shows the generation time, while the right plot shows the detection time for various watermarking models (NoWatermark, FKE, DWI, HDWI) as a function of key capacity ( K ranging from 0 to 2000).

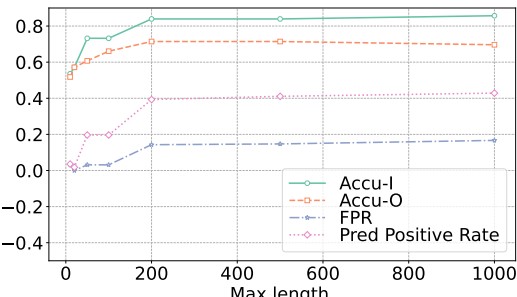

Figure 11: Sequence Length Analysis. The figure presents the impact of sequence length (ranging from 20 to 1000) on metrics (Accu-I, Accu-O, FPR, and Predicted Positive Rate) for the HDWI model with a watermark ratio $r = 0.5$.

## A.8 RUNTIME ANALYSIS

To evaluate the computational cost of the watermarking models, we conducted a runtime analysis experiment by testing the runtime for the same 100 samples across different models, including the "NoWatermark" generation. The results, shown in Appendix A.8, reveal that (1) in the generation phase, the runtime for watermarking models is slightly higher than for non-watermarked generation, as additional time is required for hashing and key encoding, (2) the runtime for all models during the generation phase is independent of the key capacity, since the watermark encoding process only runs once and does not depend on the size of the key capacity, and (3) in the detection phase, the runtime for all watermarking models increases linearly with the key capacity, as the detection process involves multiple iterations over possible keys to identify the best matching key.

## A.9 SEQUENCE LENGTH ANALYSIS

To evaluate how performance is influenced by the length of generated sequences, we conducted a sequence length analysis experiment using the HDWI model with a watermark ratio $r = 0.5$. The experiment tested sequence lengths ranging from 20 to 1000, and the results are presented in Figure 11. The following observations can be made: (1) as the sequence length increases, the accuracy scores improve, as longer sequences allow for clearer embedding of the watermark into the generated text, (2) as the sequence length grows, the FPR metric also increases; however, this does not necessarily indicate worsening false recognition problems. When the text length is short, the model rarely recognizes any sequence as watermarked, leading to accuracy scores close to 0.5 and FPR close to 0. As the sequence length increases, the predicted positive rate rises, resulting in more false positives, and (3) based on the experiment, the models begin to recognize watermarked text effectively when the token length exceeds 50, and they achieve good performance when the token length exceeds 200.

|          | Accu-I↑ | Accu-O↑ | FPR↓   | Sim↑  |          | Accu-I↑ | Accu-O↑ | FPR↓    | Sim↑  |
|----------|---------|---------|--------|-------|----------|---------|---------|---------|-------|
| FKE      | 0.883   | 0.834   | 0.175  | 0.934 | FKE      | 0.955   | 0.95    | 0.0877  | 0.901 |
| PKE      | 0.797   | 0.679   | 0.20   | 0.931 | PKE      | 0.873   | 0.80    | 0.0657  | 0.905 |
| MultiBit | 0.893   | 0.655   | 0.0825 | 0.922 | MultiBit | 0.941   | 0.732   | 0.175   | 0.89  |
| DWI      | 0.963   | 0.744   | 0.0459 | 0.932 | DWI      | 0.955   | 0.831   | 0.118   | 0.905 |
| HDWI     | 0.923   | 0.73    | 0.0517 | 0.923 | HDWI     | 0.973   | 0.863   | 0.00952 | 0.892 |
| MR       | 0.902   | 0.739   | 0.0769 | 0.923 | MR       | 0.946   | 0.846   | 0.0275  | 0.892 |
| SR       | 0.963   | 0.773   | 0.0428 | 0.923 | SR       | 0.943   | 0.832   | 0.0906  | 0.892 |

Table 5: BioASQ dataset results.      Table 6: LegalQA dataset results.

## A.10 Experiments with More Datasets

To demonstrate the applicability of our model across different scenarios, we conducted experiments on two domain-specific datasets: a biomedical question dataset, BioASQ (Krithara et al., 2023), and a legal dataset, LegalQA[3]. We evaluated our models on these datasets, and the results are presented in Table 5 and Table 6. The findings show that (1) the performance trends on these domain-specific datasets are generally consistent with those in the main experiment, with our proposed methods achieving superior results compared to other models, and (2) the similarity scores in both datasets are as high as 0.9, indicating that the watermarking method minimally alters the output text, even in highly specific domains.

## A.11 Experiments with Indication Ratio Parameter $r_d$

The indication ratio parameter $r_d$ controls the ratio between tokens used to encode the indicator variable and those used to encode key information. We conduct an experiment to evaluate how different values of $r_d$ affect the results, as shown in Figure 12. The findings are summarized as follows:

(1) As $r_d$ increases, both DWI and HDWI exhibit an improvement in the Accu-I score. This demonstrates that using more tokens to encode the indicator variable enhances the accuracy of detecting whether the text is watermarked, thereby validating the correctness of our proposed method and theoretical analysis. (2) With an increase in $r_d$, the Accu-O score initially increases and then decreases. At smaller values of $r_d$, the performance improves as more tokens are available to detect whether the text is watermarked. However, when $r_d$ becomes too large, it impairs the detection of key information, leading to a decline in overall performance. (3) DWI performs worse in both Accu-I and Accu-O when $r_d$ is small. This occurs because DWI does not reuse key information to detect whether the text is watermarked, giving HDWI an advantage at smaller $r_d$ values. This further underscores the effectiveness of the HDWI method. (4) As $r_d$ increases, the FPR decreases. Allocating more tokens to encode the indicator variable helps alleviate the false positive problem, improving overall robustness.

## A.12 Multi-bit Error Bound Analysis

Yoo et al. (2023b); Wang et al. (2024) extended Kirchenbauer et al. (2023a)'s method to support multi-bit encoding. Their approach detects if a text is watermarked by use of a binomial statistic (Yoo et al., 2023b). However, since the statistic is based on the maximal value of multiple binomial variables, it should no longer be considered a measure of a binomial distribution, but instead an approximate Gumbel distribution (Kotz & Nadarajah, 2000; Haan & Ferreira, 2006).

As the parameter for the Gumbel distribution is challenging to compute, we directly derive a novel bound for the composed extreme variable. Our analysis reveals that this method continues to suffer from the false recognition problem.

We follow the notation in Yoo et al. (2023b), and use $[r]$ to denote the sequence of length $r$, $[r] = [1, 2, \cdots, r]$. Given a generated sequence $[x_1, \cdots, x_T]$, Yoo et al. (2023b) first uses a hash key to compute the position $p_t$ of the message $m$ for the $t$-th token, denoted as $\rho_t = m[p_t]$, where $p_t \in [b]$ and $\rho_t \in [r]$. Here, $b$ is the message length, and $r$ indicates the number of bits each position encodes.

---

[3]https://huggingface.co/datasets/dzunggg/legal-qa-v1

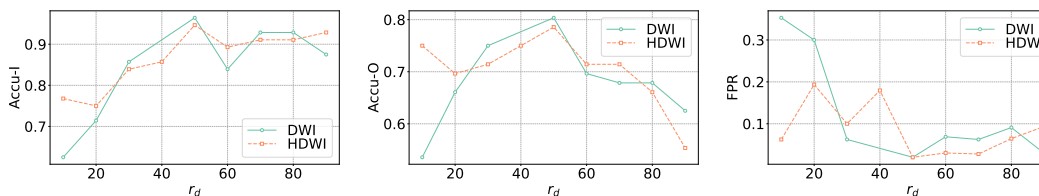

Figure 12: Experiment results with varying indication ratio parameter $r_d$. Higher $r_d$ corresponds to more tokens allocated for encoding the indicator variable.

Finally, the vocabulary $V$ is divided into $r$ blocks $[V_1, \cdots, V_r]$, and $\delta$ is added to the logits of all tokens in the $\rho_t$-th partition $V_{\rho_t}$.

When determining if a text is watermarked, the method calculates the maximal count in each vocabulary block for each position $p_t$ normalized by the total count allocated to that position. For the $p_t$-th position, the random variable $C_{p_t} \in [0, 1]$ can be denoted as:

$$C_{p_t} = \max_{\rho \in [r]} \left\{ \frac{\sum_{t=1}^{T} \mathbb{1}(x_t \in V_\rho) \cdot \mathbb{1}(p_t = \rho)}{\sum_{t=1}^{T} \mathbb{1}(p_t = \rho)} \right\}.$$

Following Yoo et al. (2023b), we approximate the distribution for each block $\rho$ using a binomial distribution. The total count for each block is approximated as $\frac{T}{b}$. Therefore, we have:

$$C_{p_t} = \max\left( \frac{X_1}{T/b}, \cdots, \frac{X_r}{T/b} \right), \quad \text{where} \quad X_\rho \sim \text{Binomial}\left( \frac{T}{b}, \frac{1}{r} \right).$$

Yoo et al. (2023b) claims that if the text is unwatermarked, $C_{p_t} \approx \frac{1}{r}$. Based on this, the detection method tests if $C_{p_t}$ exceeds a predefined threshold, classifying the text as watermarked if this is the case. However, Yoo et al. (2023b)'s approach approximates the distribution of $C_{p_t}$ with a binomial distribution. Since $C_{p_t}$ is the maximum of i.i.d. distributions, it is, in fact, a Gumbel distribution. As a result, even when the text is not watermarked, $C_{p_t}$ is still likely to exceed $\frac{1}{r}$, leading to excess false positives.

To further demonstrate this issue, we provide a theoretical analysis of how the random variable $C_{p_t}$ grows as the key capacity increases. Given the difficulty of computing the parameters of the Gumbel distribution, we further analyze its tail bounds to examine how the parameter $b$ affects $C_{p_t}$. We first present the following theorem:

**Theorem 4.** *Let* $C_{p_t} = \max\left( \frac{X_1}{T/b}, \frac{X_2}{T/b}, \cdots, \frac{X_r}{T/b} \right)$, *where* $X_\rho \sim \text{Binomial}\left( \frac{T}{b}, \frac{1}{r} \right)$ *for all* $\rho \in [r]$. *Then, the probability that* $C_{p_t}$ *exceeds a threshold* $y$ *is bounded by:*

$$\Pr(C_{p_t} \geq y) \leq r \cdot \exp\left( -\frac{2T\left( y - \frac{1}{r} \right)^2}{b} \right).$$

*Proof.* For each block $\rho \in [r]$, the normalized count is $\frac{X_\rho}{T/b}$, where $X_\rho \sim \text{Binomial}\left( \frac{T}{b}, \frac{1}{r} \right)$. The expectation of $\frac{X_\rho}{T/b}$ is:

$$\mathbb{E}\left[ \frac{X_\rho}{T/b} \right] = \frac{\mathbb{E}[X_\rho]}{T/b} = \frac{1}{r}.$$

We aim to bound the probability $\Pr\left( \frac{X_\rho}{T/b} \geq y \right)$. This is equivalent to:

$$\Pr\left( \frac{X_\rho}{T/b} \geq y \right) = \Pr\left( X_\rho \geq y \cdot \frac{T}{b} \right).$$

Using Hoeffding's inequality for $X_\rho$, we have:

$$\Pr\left( X_\rho \geq y \cdot \frac{T}{b} \right) \leq \exp\left( -\frac{2\left( y \cdot \frac{T}{b} - \mu \right)^2}{T/b} \right),$$

where $\mu = \mathbb{E}[X_\rho] = \frac{T}{b} \cdot \frac{1}{r}$.

Substitute $\mu$ into the inequality:

$$\Pr\left(X_\rho \geq y \cdot \frac{T}{b}\right) \leq \exp\left(-\frac{2\left(y \cdot \frac{T}{b} - \frac{T}{b} \cdot \frac{1}{r}\right)^2}{T/b}\right).$$

Simplify the argument of the exponential:

$$\Pr\left(\frac{X_\rho}{T/b} \geq y\right) \leq \exp\left(-\frac{2T\left(y - \frac{1}{r}\right)^2}{b}\right).$$

Now, for the maximum $C_{p_t} = \max\left(\frac{X_1}{T/b}, \frac{X_2}{T/b}, \cdots, \frac{X_r}{T/b}\right)$, we use the union bound:

$$\Pr(C_{p_t} \geq y) \leq \sum_{\rho=1}^{r} \Pr\left(\frac{X_\rho}{T/b} \geq y\right).$$

Since the bound for each $\rho$ is identical, we multiply the single block bound by $r$:

$$\Pr(C_{p_t} \geq y) \leq r \cdot \exp\left(-\frac{2T\left(y - \frac{1}{r}\right)^2}{b}\right).$$

This completes the proof. $\square$

It can be observed from Theorem 4 that as the message length $b$ increases, the probability that $C_{p_t}$ exceeds a certain threshold, $\Pr(C_{p_t} \geq y)$, also increases. This implies that as the key capacity grows, the method becomes more prone to false recognition problems. One might argue that increasing $r$ can also increase the key capacity. However, it should be noted that as $r$ increases significantly, the vocabulary will be divided into $r$ blocks, causing the "green list" to become smaller and smaller. This reduction in the green list size makes it increasingly difficult to contain feasible next tokens, further complicating the watermarking process.

We further conducted a numerical experiment to demonstrate how the distribution shifts as $b$ increases. The results are presented in Figure 13. We fix $r = 10$, indicating that each position contains 10 bits of information, and vary the message length $b \in [2, 20]$. Additionally, we plot the desired binomial distribution for $r = 10$ using the red line, as expected in the original paper. The results demonstrate that (1) as the message length $b$ increases, the expectation of the random variable $C_{p_t}$ also rises. For the origional Multibit method, the expected value is 0.1, but it continues to grow as $b$ increases, further validating the correctness of our theoretical analysis. This shift causes the unwatermarked text to resemble watermarked text, making it more challenging to distinguish them using a threshold. (2) Compared with the original binomial distribution, applying the max operation shifts the distribution to the right, resulting in a narrower distribution with reduced variance.

## A.13    WINDOW SIZE PARAMETER $h$

We evaluate whether the window size parameter significantly impacts the generation quality, and the results are presented in Figure 14. It can be observed that as the window size $h$ increases, the text quality scores remain in the range of 69 to 71. In this experiment, no substantial changes in text quality were observed as the window size $h$ varied.

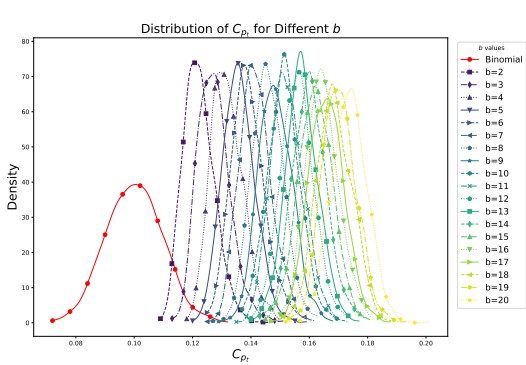

Figure 13: Distribution of $C_{p_t}$ with respect to different walues of $b$.

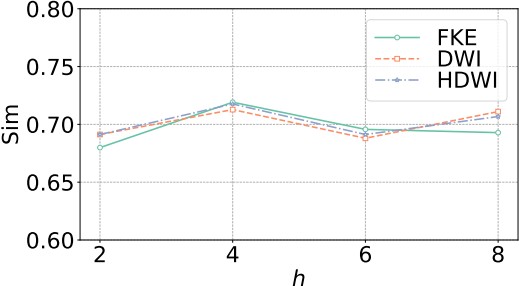

Figure 14: Text quality with respect to different window sizes $h$.

