# OpenReview forum: "Watermarking for User Identification in Large Language Models"
_ICLR.cc/2025/Conference — Submitted to ICLR 2025_

### Official Review · Reviewer_qkCQ · 2024-10-23

**Soundness:** 2
**Presentation:** 1
**Contribution:** 2
**Rating:** 5
**Confidence:** 3

**Summary:**

This paper addresses the false recognition problem in watermarking large language models (LLMs), specifically focusing on simultaneous detection of machine-generated text and user identification. The authors propose a novel Dual Watermark Inspector (DWI) method that separately encodes indicator and key information into generated text. They provide theoretical bounds for false positive rates and demonstrate empirically that their method significantly reduces false positives compared to baseline approaches. The paper includes extensive experiments validating the method's effectiveness across different settings and robustness against insertion/deletion attacks.

**Strengths:**

- Practical relevance for real-world LLM deployment and governance
- Comprehensive empirical evaluation across various settings (watermark ratios, key capacities, sample sizes)

**Weaknesses:**

While the paper presents a sound approach, it has several limitations in both technical depth and presentation clarity that could be improved upon.

1. Limited comparison with other recent watermarking methods beyond the baseline FKE approach. The paper primarily focuses on comparing with one baseline method, missing opportunities to demonstrate advantages over other state-of-the-art approaches.

2. The trade-off between accuracy and false positive rate could be analyzed more thoroughly. While the paper shows improvements in false positive rates, it lacks detailed analysis of how this affects other performance metrics across different scenarios.

3. The choice of ratio r between indicator and key information tokens appears somewhat arbitrary. The paper does not provide sufficient theoretical or empirical justification for the selected ratio values.

4. Impact on text quality and fluency not thoroughly discussed or evaluated. The paper focuses on detection metrics but lacks analysis of how the watermarking method affects the generated text's quality and naturalness.

5. Computational overhead of the proposed method compared to baseline not clearly addressed. The additional complexity introduced by the dual watermarking approach is not quantified or discussed in terms of practical implementation costs.

6. Writing and presentation issues detract from the paper's clarity. Several technical terms are used before being properly defined, figure captions lack detail, and important implementation details are scattered between the main text and appendix. The mathematical notation also shows some inconsistency across sections.

**Questions:**

- How does the method perform with very short text sequences?
- Is there a minimum length requirement?
- Could the approach be extended to handle hierarchical or structured user identification beyond simple integer keys?
- What is the impact of vocabulary size on the method's performance?
- How does the method handle adversarial attacks beyond simple insertion/deletion?

---

> ### Author Response · Authors · 2024-11-23
>
> **1.** Limited comparison with other recent watermarking methods beyond the baseline FKE approach, missing opportunities to demonstrate advantages over state-of-the-art approaches.
>
> **A:** We have added a new baseline model, Multi-bit (see the main experiments), along with a detailed theoretical analysis and numerical experiment of the false recognition problem for this model. please refer to the general response M6 and Appendix A.12.
>
>
>
> **2.** Lack of detailed analysis on the trade-off between accuracy and false positive rate across different scenarios.
>
> **A:** We have conducted such analyses in Section 5.5 and Appendix A.11, where we vary the watermarked text ratio and observe changes in Accuracy (Acco-I and Accu-O) and FPR. In Section 5.6, we analyze how increasing key capacity impacts Acco-I, Accu-O, and FPR. Appendix A.10 presents results on how Accuracy (Acco-I and Accu-O) and FPR vary across different datasets, reflecting two distinct application scenarios. In Appendix A.9, we examine the influence of sequence length on Accuracy and FPR. Additionally, in Section 5.7, we evaluate how changing the backbone model affects Accuracy and FPR. These experiments collectively demonstrate how Accuracy and FPR evolve under various scenarios.
>
> **3.** Insufficient theoretical or empirical justification for the choice of ratio $r_d$ between indicator and key information tokens.
>
> **A:** It is important to note that we have already conducted such experiments in Section 5.3. Additionally, we have included a new experiment comparing results with different $r_d$. For further details, please refer to the general response (M4).
>
>
> **4.** Limited discussion or evaluation of the impact of the watermarking method on text quality and fluency.
>
> **A:** We have incorporated a new metric, as proposed by Fernandez et al. (2023), to assess the quality of the generated text. See general response (M8) for more details.
>
>
>
> **5.** Computational overhead compared to the baseline is not quantified or discussed in terms of practical implementation costs.
>
>
> **A:** We have added a new experiment to analyze the computational cost. Please refer to the general response (M2) and Appendix A.8 for details.
>
> **6.** Writing and presentation issues, including undefined technical terms, insufficient figure captions, scattered implementation details, and inconsistent mathematical notation.
>
> **A:** We have refined the technical aspects of the paper. Specifically, we revised several term definitions and added a new algorithmic outline to clarify the procedure (see general response M9 and Algorithm 1 in the updated version). Additionally, we have provided detailed captions for all figures and tables and addressed the issue of overused mathematical notations.
>
>
> **7.** How does the method perform with very short text sequences? Is there a minimum length requirement?
>
> **A:** Generally, longer texts contain more information, which improves inference accuracy. We have added a new experiment to report performance across different text lengths. For details, please refer to the general response M3 and Appendix A.9.
>
> **8.** Could the approach be extended to handle hierarchical or structured user identification beyond simple integer keys?
>
> **A:** Technically, the method supports user keys, as any information that can be converted into a sequence can be used to convey such information. However, it should be noted that we have proven that as the key capacity increases, the false recognition problem becomes more severe (see Section 4 for theoretical analysis and Section 5.6 for experimental results). Therefore, we do not recommend using overly complex information as keys.
>
> **9.** What is the impact of vocabulary size on the method’s performance?
>
> **A:** The method’s performance is not significantly influenced by vocabulary size. For the backbone by Aaronson & Kirchner (2023), performance is independent of the vocabulary size. For the backbone by Kirchenbauer et al. (2023a), the vocabulary is divided into several partitions, but it is the number of partitions, rather than the vocabulary size itself, that affects the results. Please refer to the theoretical analysis in Appendix A.12 and Theorem 4 for further details.
>
> **10.** How does the method handle adversarial attacks beyond simple insertion/deletion?
>
> **A:** We have added a new experiment on the paraphrase attack. Please refer to the details in the general response M1.

---

> > ### Comment · Reviewer_qkCQ · 2024-12-03
> > **Thanks for your response!**
> >
> > Thank you for your response. The extra analysis have alleviated my concerns, and I have raised my score to reflect this.

---

### Official Review · Reviewer_rASn · 2024-10-31

**Soundness:** 2
**Presentation:** 1
**Contribution:** 3
**Rating:** 5
**Confidence:** 4

**Summary:**

This paper presents a new watermark encoding approach that divides the tasks of identifying whether text is generated by a large language model (LLM) and determining which user generated it. By separately encoding these two types of information, the proposed method alleviates the issue faced by traditional Full Key Encoding (FKE) methods, which experience increased false positive rates (FPR) as the number of users grows.

**Strengths:**

1. Identified a Significant Problem: The issue addressed in this paper is highly relevant and important, particularly in the context of managing LLM-generated content.
2. Effective Experimental Results: The experiments convincingly demonstrate that the proposed method effectively mitigates the false positive rate (FPR), particularly as the user count increases.

**Weaknesses:**

1. Insufficient Comparison with Existing Methods: While the authors constructed some baselines for comparison, the analysis against existing FKE methods is not comprehensive. Notably, the paper does not mention other multi-bit watermarking techniques that can extract user identity, such as MPAC[1] and Codable Watermark[2].
2. Limited Experimental Scope: The main experiments rely on only one model and one dataset, which raises concerns about the generalizability of the findings. A broader range of models and datasets would strengthen the claims made.
3. Lack of Scientific Rigor in Writing: The overall writing lacks scientific rigor, with unclear logic and weak structural organization. Improved clarity and coherence would enhance the readability and impact of the paper.

Reference:
[1] Advancing Beyond Identification: Multi-bit Watermark for Large Language Models via Position Allocation.
[2] Towards Codable Text Watermarking for Large Language Models.

**Questions:**

See the weaknesses section.

---

> ### Author Response · Authors · 2024-11-23
>
> **1.** Only compare with existing FKE methods, lack of mention of multi-bit watermarking techniques.
>
> **A:** For multi-bit watermarking, we have added a new theoretical analysis, numerical experiments, and baseline models to the main experiment. For more details, please refer to the general response M6 and Appendix A.12.
>
> **2.** Limited experimental scope, relying on only one model and one dataset, raising concerns about generalizability.
>
> **A:** We have used two backbone models in our experiments. The main experiments are based on one model, while we also conducted experiments using the model proposed by Aaronson & Kirchner (2023) in the main experiments and Kirchenbauer et al. (2023a) in Section 5.7.
>
> Additionally, we incorporated two datasets from the biomedical and legal domains, with detailed descriptions provided in the general response M5 and Appendix A.10.
>
>
> **3.** Lack of scientific rigor in writing, with unclear logic and weak structural organization.
>
> **A:** We have further refined the paper and added a new algorithmic outline in Algorithm 1, making the procedure clearer.

---

> > ### Comment · Reviewer_rASn · 2024-12-03
> >
> > Thank you for providing the analysis of the multi-bit watermarking algorithm. I believe that adding this analysis enhances the comparison with existing work, making it more comprehensive. The additional experiments have alleviated my concerns, and I therefore feel it is appropriate to raise the score to reflect this.

---

### Official Review · Reviewer_QsKM · 2024-11-04

**Soundness:** 2
**Presentation:** 1
**Contribution:** 2
**Rating:** 5
**Confidence:** 4

**Summary:**

This paper explores a new problem for LLM watermarking with a specific focus on user identification. The authors address the challenge of false recognition, where non-watermarked text may be mistakenly classified as watermarked, especially as the user count increases. To tackle this, the paper introduces a Dual Watermark Inspector (DWI) and a Hybrid Dual Watermark Inspector (HDWI) approach, aiming to improve the robustness of watermark detection by reducing false positives. The paper offers a combination of theoretical derivations and empirical validations, suggesting that DWI and HDWI outperform existing watermarking methods in accuracy and false positive control.

**Strengths:**

1. Addresses a pressing issue of false positives in multi-user watermarking, which is important for the practical use of watermarking in AI applications.

2. Introduces a new dual watermarking approach that embeds user identity while preserving the text's integrity, showing significant improvement in error rates.

**Weaknesses:**

1. The presentation lacks clarity in explaining the watermarking process. I suggest that the authors include a detailed algorithmic outline to demonstrate how watermark generation and detection are performed. It is currently unclear how the integer key (user ID) is embedded into the generated text. Additionally, based on my understanding, this method is an extension involving when to add Gumbel noise for embedding the watermark, with the primary approach building on existing methods.

2. The citation formatting (using \citep and \citet) should be corrected. Furthermore, the term "Stochastic sampling" in lines 151-152 needs clarification—what exactly is meant by this here? Regarding the hash function, you currently rely on the previous two tokens for hashing. However, as Aaronson suggests, a small window size can compromise pseudorandomness. It would be beneficial to either use a larger window size or store a two-gram history to prevent repetition issues.

3. From my understanding, user identification could naturally be addressed through multi-bit watermarking techniques. The statement in lines 104-106, "no methods have explored the task of retrieving information from watermarked text intermingled with unwatermarked text," is unclear. I recommend expanding on this point and providing a more detailed discussion of multi-bit watermarking, especially in relation to user identification.

**Questions:**

How do DWI and HDWI methods perform computationally when applied to LLMs with larger user bases?

---

> ### Author Response · Authors · 2024-11-23
>
> **1.** Could the authors include a detailed algorithmic outline to demonstrate how watermark generation and detection are performed?
>
> **A:** Thank you for the suggestion. We have included the algorithmic outline in Algorithm 1. Please refer to the general response M9 and our updated version for further details.
>
> **2.** How is the integer key (user ID) embedded into the generated text?
>
> **A:** The integer key (user ID) is embedded into the generated text through a controlled sampling process using a hash function and the Gumbel-Max trick. A hash key derived from the previous two tokens determines whether the current token encodes a watermark indicator or key information. If encoding key information, the user ID is incorporated as a salt key, which is then used to adjust the logits via Gumbel noise. This process subtly influences token selection, embedding the user ID into the text while maintaining robustness and preserving text diversity. Please refer to the general response M9 and our updated version for further details.
>
> **3.** The citation formatting (using \citep and \citet) should be corrected.
>
> **A:** Thank you for pointing this out. We have revised all the citations accordingly. Please refer to our updated version.
>
>
> **4.** What is meant by “Stochastic sampling” in lines 151-152, and could it be clarified?
>
> **A:** “Stochastic sampling” refers to the process of selecting the next token in text generation based on a probability distribution derived from the model’s logits. Instead of deterministically choosing the token with the highest probability (e.g., greedy decoding), stochastic sampling randomly selects a token, with the likelihood of selection proportional to its predicted probability. This approach introduces variability into the output, promoting diversity and creativity in the generated text while remaining guided by the underlying distribution. We have included a more detailed explanation in the main text.
>
> **5.** Could the authors address concerns about using a small window size for hashing, as suggested by Aaronson, and consider alternatives like using a larger window size or storing a two-gram history to prevent repetition issues?
>
> **A:** Using a two-gram window size is a de facto choice, as suggested by Fernandez et al. (2023), for setting the window size. Increasing the window size may make the method more vulnerable to adversarial attacks. Furthermore, we have not observed significant repetition issues in our generated text, so we have opted to retain this setting without modifications.
>
>
> **6.** Expand on providing a more detailed discussion of multi-bit watermarking, especially in relation to user identification.
>
> **A:** We have added a new theoretical analysis, numerical experiments, and baseline models to the main experiment. For more details, please refer to the general response M6 and Appendix A.12.
>
>
> **7.** Performance computationally when applied to LLMs with larger user bases.
>
> **A:** These results are presented in Table 3 and Figure 5, with further details in Section 5.6. The experiments show that as key capacity $K$ increases, the FKE method’s performance decreases significantly due to its reliance on the maximal score across the key space, consistent with our theory. In contrast, the HDWI method remains robust, as the indicator variable ensures that the false positive rate (FPR) is unaffected by the total key count, demonstrating scalability for larger user bases.

---

> > ### Comment · Reviewer_QsKM · 2024-11-25
> >
> > > A: “Stochastic sampling” refers to the process of selecting the next token in text generation based on a probability distribution derived from the model’s logits. Instead of deterministically choosing the token with the highest probability (e.g., greedy decoding), stochastic sampling randomly selects a token, with the likelihood of selection proportional to its predicted probability. This approach introduces variability into the output, promoting diversity and creativity in the generated text while remaining guided by the underlying distribution. We have included a more detailed explanation in the main text.
> >
> > You mentioned Top-k sampling (Fan et al., 2018), Nucleus sampling (Holtzman et al., 2020), and Stochastic sampling (Fan et al., 2018; Holtzman et al., 2020; Fu et al., 2021). I assumed that Top-k and Top-p sampling are both Stochastic sampling methods.
> >
> > > A: Using a two-gram window size is a de facto choice, as suggested by Fernandez et al. (2023), for setting the window size. Increasing the window size may make the method more vulnerable to adversarial attacks. Furthermore, we have not observed significant repetition issues in our generated text, so we have opted to retain this setting without modifications.
> >
> > Using a small window size can also make the method problematic. For example, with the same two-gram prefix, the randomness becomes fixed, which may lead to frequent repetitions or reduced detectability.

---

> > > ### Author Response · Authors · 2024-11-26
> > >
> > > Thank you for the follow-up questions. The answers are as follows.
> > >
> > > **1.** Top-k and Top-p sampling are both Stochastic sampling methods.
> > >
> > > **A:** No. Here, by stochastic sampling, we refer to sampling from the entire vocabulary, excluding Top-k or Top-p strategies for simplicity. However, this approach can be readily extended to include Top-k and Top-p sampling, as the probabilities involved in detection remain consistent.
> > >
> > > **2.** Using a small window size can also make the method problematic. For example, with the same two-gram prefix, the randomness becomes fixed, which may lead to frequent repetitions or reduced detectability.
> > >
> > > **A:** Thank you for pointing this out. This parameter can be easily adjusted to larger values without requiring changes to other parts. We have modified the text to clarify for readers that this parameter can be set to different values to avoid confusion. The setting of the window length is unrelated to our proposed methods and, therefore, does not affect the conclusions.
> > >
> > > It is worth noting that in our experiments, we did not observe extensive repetition under the current settings, and the generated text quality is acceptable (as evidenced by the newly added Sim score). The primary focus of this paper is not on addressing repetition, and this parameter setting proves to be effective for our purposes. To further demonstrate this, we conducted an additional experiment detailed in A.13. Our results indicate that while changes in window size influence text quality, the variations are not drastic. Consequently, this setting is deemed appropriate for our use to test our methods.

---

### Official Review · Reviewer_e57V · 2024-11-12

**Soundness:** 2
**Presentation:** 2
**Contribution:** 2
**Rating:** 6
**Confidence:** 2

**Summary:**

The paper proposes a method for achieving both output watermarking and user identification for LLMs.

**Strengths:**

- LLM watermarking is an important topic for the research community.
- The inclusion of a theoretical analysis strengthens the paper.

**Weaknesses:**

- Although the proposed method improves on baseline FPR values, the overall FPR (>0.01) is still too high for practical applications.
- Watermarks can significantly impact text quality, but the paper lacks any analysis of this issue.
- Paraphrasing, whether done once or multiple times, may weaken the watermark because of the hash's sensitivity to small changes.

Minor Points
- The citations need to be adjusted. The authors should use \citep for most citations, except where the citation is the subject or object of the sentence.
- Some tables and figures would benefit from more descriptive captions. For example, Figures 2–7 and Tables 2–5 lack clear descriptions.

**Questions:**

Given a fixed prompt, does the watermark affect output diversity?

---

> ### Author Response · Authors · 2024-11-23
> **Author Response**
>
> **1.** While the proposed method improves baseline FPR values, the overall FPR (>0.01) is still too high for practical applications.
>
> **A:** We respectfully disagree with the assertion that an FPR greater than 0.01 renders the method impractical. The acceptability of the FPR threshold is highly application-dependent and varies based on the specific requirements and constraints of the task. Additionally, achieving an FPR below 0.01 is a challenging goal for most applicable methods in this domain. We envision this method as a practical solution for initial filtering of suspicious text, which can then be forwarded for further manual or automated verification.
>
> **2.** The paper lacks any analysis of how watermarks impact text quality.
>
> **A:** We have incorporated a new metric, following Fernandez et al. (2023), to assess the quality of the generated text. See general response (M8) for more details.
>
> **3.** Does paraphrasing, whether done once or multiple times, weaken the watermark due to the hash’s sensitivity to small changes?
>
> **A:** We have included a new experiment to evaluate the method’s performance under paraphrase attacks. Please refer to the general response (M1) for further details.
>
>
> **4.** Should citations be adjusted to use \citep except when the citation is the subject or object of a sentence?
>
> **A:** Thank you for pointing this out. We have revised all the citations accordingly. Please refer to our updated version.
>
>
> **5.** Figures 2–7 and Tables 2–5 require more descriptive captions.
>
> **A:** Thank you for pointing this out. We have revised all the Figures and Tables accordingly. Please refer to our updated version.
>
> **6.** Given a fixed prompt, does the watermark affect output diversity?
>
> **A:** The watermark itself does not affect output diversity. This is because, after applying the Gumbel trick, the process becomes equivalent to a sampling operation, which can preserve diversity as long as the randomness is appropriately controlled. It is important to note that the source of randomness originates from the uniform distribution prior to applying the Gumbel trick.

---

> > ### Comment · Reviewer_e57V · 2024-12-03
> >
> > Thank you for your rebuttal efforts. I’ve reviewed them and updated my score accordingly.

---

### Author Response · Authors · 2024-11-23
**General Response (1/2)**

We thank the reviewers for their valuable comments. To address these issues, we have made several improvements to the paper. Specifically, we have added experiments on rephrasing attacks (M1), runtime analysis (M2), sequence length analysis(M3), indication ratio parameter $r_d$ analysis (M4). Additionally, we have conducted new experiments on two datasets, BioASQ and LegalQA (M5). A new theoretical analysis (M6) for multi-bit methods based on a dictionary color list has also been added, along with numerical experiments to further support the theoretical findings. We have also introduced a new baseline model (M7) in the experimental section and incorporated a new metric (M8) to evaluate text generation quality. We also add a new algorithmic outline. The detailed modifications are provided below, and we will address each question under the corresponding reviewer comments.

**M1. We have added experiments on rephrasing attacks. (Details in Appendix A.7)** Specifically, we conducted a paraphrase attack to evaluate the robustness of the proposed methods. We employed the Parrot_Paraphraser toolkit to rephrase sentences with watermarks and tested the ability of our models to detect the watermark and key information. The results demonstrate that (1) our proposed DWI and HDWI models still outperform the FKE method aster the rephrasing attack, (2) although detection accuracy decreases post-paraphrasing, it remains above 0.5, indicating that the methods can still identify watermarked text and associated keys.

**M2. We have added new experiments on runtime analysis. (Details in Appendix A.8)** We evaluated the computational cost of the watermarking models by testing the runtime across various models, including “NoWatermark” generation. Results show that (1) watermarking models slightly increase generation runtime due to hashing and key encoding, (2) generation runtime is independent of key capacity, as encoding occurs only once, and (3) detection runtime increases linearly with key capacity, as it requires iterating over potential keys.

**M3. We have added new experiments on sequence length. (Details in Appendix A.9.)** Using the HDWI model with a watermark ratio of $r=0.5$, we analyzed the impact of sequence length, ranging from 20 to 1000 tokens. Results indicate that (1) accuracy improves with longer sequences as they better embed watermarks, and (2) models effectively recognize watermarked text when token lengths exceed 50 and achieve good performance beyond 200 tokens.

**M4. We have added new experiments on the indication ratio parameter $r_d$. (Details in Appendix A.11.)** In response to the reviewer's suggestion, we have conducted additional experiments analyzing the impact of $r_d$ on performance metrics such as Accu-I, Accu-O, and FPR. The results, shown in Appendix A.11, indicate that as $r_d$ increases, both DWI and HDWI improve in Accu-I, validating our theoretical claims. However, excessively large $r_d$ values reduce Accu-O by hindering key information detection, demonstrating a tradeoff. Additionally, HDWI outperforms DWI in low $r_d$ scenarios, and increasing $r_d$ reduces FPR by mitigating false positives. This analysis highlights the robustness of our method and addresses the reviewer's concern.


**M5. We have added new experiments on two new datasets. (Details in Appendix A.10.)** To evaluate the model’s applicability across different domains, we tested it on BioASQ (biomedical questions) and LegalQA (legal texts). Results show (1) consistent performance trends with the main experiments, where our methods outperform others, and (2) high similarity scores (up to 0.9), indicating minimal alteration of output text even in domain-specific contexts.

---

> ### Author Response · Authors · 2024-11-23
> **General Response (2/2)**
>
> **M6. In-Depth Theoretical and Numerical Analysis of the Multi-Bit Encoding Method (Details in Section A.12.)**
>
> We appreciate the reviewer’s comments and have expanded our analysis of the multi-bit encoding method. Specifically, we conducted a new theoretical analysis to demonstrate that the multi-bit method also suffers from the false recognition problem as key capacity increases. Key points are summarized as follows:
>
> (1) Distribution Analysis and Novel Bound:
> We identified that the detection method in the multi-bit encoding aligns more closely with a Gumbel distribution than the originally claimed binomial distribution. Based on this observation, we derived a novel bound for the composed extreme variable as follows:
>
> Let $C_{p_t} = \max\left(\frac{X_1}{T/b}, \frac{X_2}{T/b}, \cdots, \frac{X_r}{T/b}\right)$, where $X_\rho \sim \text{Binomial}\left(\frac{T}{b}, \frac{1}{r}\right)$ for all $\rho \in [r]$. Then, the probability that $C_{p_t}$ exceeds a threshold $y$ is bounded by:
> $$\Pr(C_{p_t} \geq y) \leq r \cdot \exp\left(-\frac{2 T \left(y - \frac{1}{r}\right)^2}{b}\right).$$
>
> This bound reveals that as the message length $b$ increases, the probability of false positives $\Pr(C_{p_t} \geq y)$ also increases, exacerbating the false recognition problem.
>
> (2) Tail Bound Analysis and Implications:
> To address challenges in computing Gumbel parameters, we analyzed its tail bounds and showed that the detection threshold increases with key capacity. This growth causes unwatermarked text to increasingly resemble watermarked text.
>
> (3) Numerical Validation (Fig. 13):
> Numerical experiments validated our theoretical findings. Specifically, as \( b \) increases, the expected detection statistic rises, the distribution narrows, and variance decreases. These effects complicate accurate classification, confirming that false recognition becomes more severe as key capacity increases.
>
> These theoretical and experimental results support our main paper’s claim, emphasizing the limitations of the multi-bit watermarking method and providing a deeper understanding of its inherent challenges.
>
> **M7. We have added new baseline models (details in the Experiments section).** To enhance the comparison of multi-bit methods, we incorporated them as new baseline models, as detailed in the updated Experiments section. It demonstrates that the multi-bit model applied in this scenario is capable of encoding key information and indicator variables. However, its performance is inferior to our proposed methods, highlighting the effectiveness of our proposed framework.
>
>
> **M8. We have added a new similarity metric, Sim (details in the Experiments section).** To assess the quality of generated text after watermarking, we adopt the approach of Fernandez et al. (2023), utilizing cosine similarity (Sim) between watermarked and unwatermarked text to evaluate generation quality and measure information loss. The experimental results indicate that the sSim scores of all watermarking methods are at a similar level, demonstrating that our method does not compromise quality compared to other watermarking methods.
>
>
> **M9. We have added a new algorithmic outline to make the procedure clearer.** (Details in Algorithm 1)
> We have included a new algorithmic outline illustrating the generation and detection processes to enhance clarity. This visual representation provides a step-by-step depiction of the procedure, making it more intuitive for readers to follow.

---

### Meta-Review · Area_Chair_NWJk · 2024-12-16

**Metareview:**

This paper addresses the challenge of false recognition in watermarking for large language models (LLMs), particularly in detecting machine-generated text and identifying specific users. The authors propose the Dual Watermark Inspector (DWI) and Hybrid Dual Watermark Inspector (HDWI) methods, which encode text generation and user identification information separately. This approach reduces false positive rates (FPR), a common issue in traditional Full Key Encoding (FKE) methods, especially as the number of users increases. Theoretical analyses and empirical experiments demonstrate the superiority of DWI and HDWI in accuracy and robustness, including resistance to insertion and deletion attacks, compared to baseline watermarking techniques.

The proposed method includes rigorous theoretical analysis supported by experiments. The authors have responded to the reviewers' concerns. Unfortunately, the revised average score remains below the acceptance threshold. Since LLM watermarking research is a hot topic in ML and there are numerous existing methods, it would be beneficial to include more discussion and, if possible, comparisons to these methods. This would make the paper stronger, in my opinion. Therefore, I encourage the authors to revise the paper based on the reviewers' comments and resubmit it to a future venue.

**Additional Comments On Reviewer Discussion:**

The author responded to the reviewer's concern and the reviewer raised the score based on the rebuttal. However, the updated score is still below the borderline. This means that non of the reviewers are not excited about the work. Thus, I recommend rejection for the paper.

---

### Decision · Program_Chairs · 2025-01-22

Reject